# Optimality in Mean Estimation:
# Beyond Worst-Case, Beyond Sub-Gaussian,
# and Beyond $1 + \alpha$ Moments

**Trung Dang**
The University of Texas at Austin
dddtrung@cs.utexas.edu

**Jasper C.H. Lee**
University of Wisconsin-Madison
jasper.lee@wisc.edu

**Maoyuan Song**
Purdue University
maoyuanrs@gmail.com

**Paul Valiant**
Purdue University
pvaliant@gmail.com

## Abstract

There is growing interest in improving our algorithmic understanding of fundamental statistical problems such as mean estimation, driven by the goal of understanding the fundamental limits of what we can extract from limited and valuable data. The state of the art results for mean estimation in $\mathbb{R}$ are 1) the optimal sub-Gaussian mean estimator by [Lee and Valiant, 2022], attaining the optimal sub-Gaussian error constant for all distributions with finite but unknown variance, and 2) the analysis of the median-of-means algorithm by [Bubeck, Cesa-Bianchi and Lugosi, 2013] and a matching lower bound by [Devroye, Lerasle, Lugosi, and Oliveira, 2016], characterizing the big-O optimal errors for distributions that have tails heavy enough that only a $1 + \alpha$ moment exists for some $\alpha \in (0, 1)$. Both of these results, however, are optimal only in the worst case. Motivated by the recent effort in the community to go "beyond the worst-case analysis" of algorithms, we initiate the fine-grained study of the mean estimation problem: Is it possible for algorithms to leverage *beneficial* features/quirks of their input distribution to *beat* the sub-Gaussian rate, without explicit knowledge of these features?

We resolve this question, finding an unexpectedly nuanced answer: "Yes in limited regimes, but in general no". Given a distribution $p$, assuming *only* that it has a finite mean and absent any additional assumptions, we show how to construct a distribution $q_{n,\delta}$ such that the means of $p$ and $q$ are well-separated, yet $p$ and $q$ are impossible to distinguish with $n$ samples with probability $1 - \delta$, and $q$ further preserves the finiteness of moments of $p$. Moreover, the variance of $q$ is at most twice the variance of $p$ if it exists. The main consequence of our result is that, no reasonable estimator can asymptotically achieve better than the sub-Gaussian error rate for any distribution, up to constant factors, which matches the worst-case result of [Lee and Valiant, 2022]. More generally, we introduce a new definitional framework to analyze the fine-grained optimality of algorithms, which we call "neighborhood optimality", interpolating between the unattainably strong "instance optimality" and the trivially weak admissibility/Pareto optimality definitions. As an application of the new framework, we show that the median-of-means algorithm is neighborhood optimal, up to constant factors. It is an open question to find a neighborhood-optimal estimator *without* constant factor slackness.

37th Conference on Neural Information Processing Systems (NeurIPS 2023).

# 1 Introduction

Mean estimation over $\mathbb{R}$ is one of the most fundamental problems in statistics: given $n$ i.i.d. samples from some unknown distribution $p$, how do we most accurately estimate the mean of $p$, with probability $\geq 1 - \delta$, from the $n$ samples? The conventional approach is to take the sample mean, the empirical average of the samples, as the estimate; this is justified by the law of large numbers, which says that that in the limit of having infinitely many samples, the sample mean will converge to the true mean. However, it has been long known that while the sample mean is optimal for estimating the mean of well-behaved distributions such as Gaussians, it is sensitive to the presence of *outliers* in samples drawn from heavy-tailed distributions, for which the sample mean estimator can have abysmal performance.

The classic median-of-means estimator [e.g., NY83; JVV86; AMS99], independently invented several times in the literature, was proposed to mitigate the sensitivity of the sample mean. Its accuracy on any distribution $p$ with finite variance is—up to constant factors—as good as the accuracy of the sample mean on a Gaussian with the same mean and variance as $p$. In other words, the error is of *sub-Gaussian rate*. The past decade has seen renewed interest across computer science and statistics in understanding the limits of the mean estimation problem. In one dimension, the current state of the art is 1) for distributions $p$ with finite variance $\sigma_p^2$, the recent mean estimator proposed by Lee and Valiant [LV22] has sub-Gaussian rate $\sigma_p \cdot (\sqrt{2} + o(1))\sqrt{\log \frac{1}{\delta}/n}$, which is tight even in its constants, up to a $1 + o(1)$ factor, and 2) the work of Bubeck, Cesa-Bianchi and Lugosi [BCL13] showing upper bounds matching the lower bounds of Devroye, Lerasle, Lugosi and Oliveira [DLLO16], which show that for heavy-tailed distributions having only a $1 + \alpha^{\text{th}}$ moment for some $\alpha \in (0, 1)$, the median-of-means algorithm in fact still achieves the optimal accuracy up to a constant factor. Both of these results, however, are optimal only in the worst case, meaning that [LV22; DLLO16] have optimal performance with respect to the class of distributions with the same $2^{\text{nd}}$ or $1 + \alpha^{\text{th}}$ moments respectively; but estimators may do better than these bounds on particular input distributions.

The natural and immediate next question is, even though Gaussian distributions are the hardest case in mean estimation, and thus sub-Gaussian performance is worst-case optimal: can one do better, at least for *some* "easier" distributions? Can we develop "instance-dependent" algorithms and analysis techniques? Is it possible for algorithms to leverage *beneficial* features/quirks of their input distribution to *beat* the sub-Gaussian rate, without explicit knowledge of these features and without losing robustness to heavy-tails?

We resolve this question and show an unexpectedly nuanced answer: "Yes in limited regimes, but in general no". For some distributions, even median-of-means can beat the sub-Gaussian bound, but only for a limited parameter regime per distribution—namely, if the number of samples is not too large (Proposition 14). In general however, we show a strong and comprehensive negative result. Our main technical result (Theorem 2) is a fine-grained indistinguishability construction: given a distribution $p$, assuming *only* that $p$ has a finite mean and absent any further assumptions, we show how to construct a distribution $q_{n,\delta}$—in terms of a sample complexity $n$ and a failure probability $\delta$—such that $p$ and $q$ are impossible to distinguish with $n$ samples, with probability $\geq 1 - \delta$, and yet the means of $p$ and $q$ are well-separated by some function $\epsilon_{n,\delta}(p)$ (stated formally in Definition 1). This in particular implies that no $n$-sample estimator with failure rate $\delta$ can simultaneously have error less than $\frac{1}{2}\epsilon_{n,\delta}(p)$ on both $p$ and $q$. The function $\epsilon_{n,\delta}(p)$ is such that, for $p$ with a finite variance, if we take $\log \frac{1}{\delta}/n \to 0$, we have $\epsilon_{n,\delta}(p) \to \Omega(\sigma_p\sqrt{\log \frac{1}{\delta}/n})$, showing that no estimator can asymptotically outperform the sub-Gaussian rate for both $p$ and $q_{n,\delta}$ simultaneously. Additionally, as shown in Section 2.1, the construction of $q_{n,\delta}$ is conservative, such that $\frac{\mathrm{d}q}{\mathrm{d}p} \leq 2$ at all points, meaning that $q$ has a finite $1 + \alpha^{\text{th}}$ moment whenever $p$ does, and furthermore, $\sigma_q^2 \leq 2\sigma_p^2$ whenever $\sigma_p^2$ exists. Thus, the same indistinguishability result still applies even if we further require the existence of higher moments for both $p$ and $q$.

The key message of our paper is that such lower bounds are to be *circumvented*, through identifying additional favorable distribution structure for the mean estimation problem.[1] This observation has already led to further work in the area, guiding the design of a new algorithm that outperforms

---

[1] "Favorable distribution structure", in particular, *cannot* mean just the existence of higher moments, since our indistinguishability construction applies even in this case.

the sub-Gaussian rate via a symmetry assumption. Gupta et al. [GLP23] show that, assuming the distribution is *symmetric* about its mean, one can achieve finite-sample and instance-dependent *Fisher information* rates for mean estimation, which can be significantly better than sub-Gaussian rates. We view our paper and the [GLP23] result together as a **call to arms** to explore other structural assumptions that can sidestep our lower bound construction.

Beyond the asymptotic implications in the finite variance setting, our results fully characterize mean estimation in the regimes of 1) finite variance and finite samples, and 2) infinite variance, and indeed, even when no $1 + \alpha^{\text{th}}$ moment exists for any constant $\alpha > 0$. In particular, we give a simple, yet very general re-analysis of the median-of-means estimator on distributions only assumed to have a finite mean. Its estimation error on distribution $p$, with probability $1 - \delta$ over $n$ samples, is $O(\epsilon_{n,\delta}(p))$ (Proposition 14), matching our main indistinguishability result up to constants (Theorem 2).

## 1.1 Our Results and Techniques

Our main result is an indistinguishability result, for every distribution $p$, which serves as a mean estimation lower bound. Ideally, given the motivation earlier in the introduction, we want to show that the sub-Gaussian rate is a lower bound, but such a bound cannot hold in finite samples. To see this, consider a distribution which has $\ll 1/n$ mass that is extremely far away from the mean, and which contributes the majority of the variance, yet only a minuscule portion of the mean. Given only $n$ samples, with high probability we will not see any samples from this mass, and so mean estimation can in fact be "effectively" performed on the conditional distribution without this outlier mass. The conditional distribution has essentially the same mean as the original, yet has far smaller variance, thus allowing us to *beat* the (variance-dependent) sub-Gaussian rate. We thus start this section by defining an error function $\epsilon_{n,\delta}(p)$ for each distribution $p$ (Definition 1), capturing the estimation error we expect for $p$, taking into account that we intuitively expect both algorithms and lower bounds to ignore outlier mass. Our main result will then construct, for every distribution $p$, a new distribution $q$ that is indistinguishable from $p$ using $n$ samples, yet has mean difference at least $\epsilon_{n,\delta}(p)$ from $p$.

**Definition 1** *Given a (continuous) distribution $p$ with mean $\mu_p$ and a real number $t \in [0, 1]$, define the $t$-trimming operation on $p$ as follows: select a radius $r$ such that the probability mass in $[\mu_p - r, \mu_p + r]$ equals $1 - t$; then, return the distribution $p$ **conditioned** on lying in $[\mu_p - r, \mu_p + r]$.*

*Given $n$ and $\delta$, we define a standard trimmed distribution $p_{n,\delta}^*$ to be the $\frac{0.45}{n} \log \frac{1}{\delta}$-trimmed version of $p$. When $\delta$ is implicit, we may denote this as $p_n^*$. We also define the error function $\epsilon_{n,\delta}(p) = |\mu_p - \mu_{p_n^*}| + \sigma_{p_n^*} \sqrt{\frac{4.5 \log \frac{1}{\delta}}{n}}$.*

Theorem 2 shows that, given any distribution $p$ with a finite mean, it is possible to construct a distribution $q_{n,\delta}$ such that 1) $p$ and $q$ are indistinguishable under $n$ samples with probability $1 - \delta$, yet 2) the means of $p$ and $q$ are separated by $\Omega(\epsilon_{n,\delta}(p))$. The construction of $q_{n,\delta}$ is also such that $\mathrm{d}q/\mathrm{d}p \leq 2$, showing that, in many senses, "$q$ does not have more extreme tails than $p$".

**Theorem 2** *Let $n$ be the sample complexity and $\delta$ be the failure probability, and recall the definition of the error function $\epsilon_{n,\delta}$ from Definition 1. Assume that there is a sufficiently small constant which upper bounds both $\frac{\log \frac{1}{\delta}}{n}$ and $\delta$. Then for any distribution $p$ with a finite mean $\mu_p$, there exists a distribution $q \neq p$ with mean $\mu_q$ such that:*

- $|\mu_q - \mu_p| \geq \frac{1}{32} \epsilon_{n,\delta}(p)$

- $\log(1 - d_{\mathrm{H}}^2(p, q)) \geq \frac{1}{2n} \log 4\delta$

- $\frac{\mathrm{d}q}{\mathrm{d}p} \leq 2$.

*In particular, by a standard fact (Fact 1) on the squared Hellinger distance, this implies that $p$ and $q$ are indistinguishable using $n$ samples, with probability $1 - \delta$. Furthermore, since $\frac{\mathrm{d}q}{\mathrm{d}p} \leq 2$, we have $\sigma_q^2 \leq \mathbb{E}_q[(X - \mu_q)^2] \leq \mathbb{E}_q[(X - \mu_p)^2] \leq 2 \mathbb{E}_p[(X - \mu_p)^2] = 2\sigma_p^2$.*

Using a standard testing-to-estimation reduction, it follows from the main theorem that there can be no estimator that can achieve error at most $\frac{1}{64} \epsilon_{n,\delta}(p)$ simultaneously on $p$ and $q_{n,\delta}$.

**Corollary 3** *Let $n$ be the sample complexity and $\delta$ be the failure probability, and recall the definition of the error function $\epsilon_{n,\delta}$ from Definition 1. Given a distribution $p$ with finite mean, consider the construction of $q$ in Theorem 2. Then, there is no estimator that achieves error less than $\frac{1}{64}\epsilon_{n,\delta}(p)$ over $n$ samples with probability $1 - \delta$, for both $p$ and $q$.*

We contrast this lower bound with more standard impossibility results that have the flavor: "one cannot estimate the mean to within $\sigma \cdot (\sqrt{2} + o(1))\sqrt{\log \frac{1}{\delta}/n}$, since there are a pair of Gaussian distributions of variance $\sigma$, separated by twice this distance, that are indistinguishable in $n$ samples up to probability $1 - \delta$". As opposed to showing the indistinguishability of "nice" distributions like Gaussians, we instead show that for *any* distribution $p$ of interest (with finite mean), we exhibit a generic construction of a hard-to-distinguish "partner" distribution $q$, of rather different mean, yet all of whose tails are comparable to those of $p$.

For reference, the definition of Hellinger distance and the standard fact we reference above are:

**Fact 1** *Consider the squared Hellinger distance between two distributions $p$ and $q$, defined as*

$$d_{\mathrm{H}}^2(p, q) = \frac{1}{2} \int (\sqrt{\mathrm{d}p} - \sqrt{\mathrm{d}q})^2$$

*If the two distributions $p$ and $q$ are such that $\log(1 - d_{\mathrm{H}}^2(p, q)) \geq \frac{1}{2n} \log 4\delta$, then there is no test that distinguishes $p$ and $q$ with probability $1 - \delta$ using $n$ samples.*

Complementing the specific construction of $q$ and analysis of its properties, we also introduce a new and general *definition* framework, speaking to the challenge of capturing "beyond worst case analysis" in this nuanced setting. In Section 3 we motivate and introduce this notion. Broadly, we want to capture the intuition of "instance-optimal" algorithms, namely, an algorithm that performs as well on the given distribution $p$ as *any* algorithm customized towards $p$; however, this definition is unattainably strong in our setting. By contrast, weaker notions such as "admissibility" or Pareto-efficiency are too weak to rule out trivial and effectively useless estimators. We introduce a new notion which we call "neighborhood optimality" that subtly blends between these notions. See Section 3 for details. Appendix A.3 also gives an in-depth discussion comparing neighborhood optimality with the more commonly-used notion of *local minimax* [AD20b; AD20a; HLY21]. While the two definitions look superficially similar, we show in the appendix—using a general proposition and a concrete example—that our new notion is a stronger and also more robust definition in the context of mean estimation.

As an application of this new framework, we show in Section 4 that the standard median-of-means estimator is neighborhood optimal up to constant factors. It is an open question to design a neighborhood-optimal estimator which does not have such constant factor slackness.

## 1.2 Open Questions

We briefly discuss a few open questions and future research directions raised by our results.

**Optimal constants for neighborhood optimality estimators—in theory and in practice**  While median-of-means enjoys neighborhood optimality as we show, the hidden multiplicative constants constants in our analysis are not tight. The median-of-means estimator is also not recommended in practice due to its large asymptotic variance [Min19]. The goal thus is to show that more modern estimators, for example [LV22], also enjoy neighborhood optimality. Ideally, such analysis would yield optimal constants.

**Other distributional structures for avoiding asymptotic sub-Gaussian lower bound**  As mentioned in the introduction, we view our paper as a *call to arms* to investigate distributional structures and assumptions that allow mean estimators to go beyond the sub-Gaussian error rate. The paper of Gupta et al. [GLP23] is the first work aiming to circumvent our lower bounds—showing the benefit of *symmetric* distributions for mean estimation. The hope, and challenge, is to find other realistic settings that enable mean estimation algorithms beyond the sub-Gaussian benchmark.

**High-Dimensional Neighborhood Optimality**  Generalizing the framework and results of this paper to the *high-dimensional* setting is a compelling direction for future work. In high dimensions,

for a distribution with covariance $\Sigma$, the corresponding sub-Gaussian rate for mean estimation under the $\ell_2$ norm is $\Theta(\frac{1}{\sqrt{n}}\sqrt{\text{tr}(\Sigma) + \|\Sigma\|\log\frac{1}{\delta}})$. We conjecture that, like in the 1-dimensional case, mean estimators cannot outperform the sub-Gaussian rate in the asymptotic regime as $n \to \infty$. More specifically, we conjecture that whenever a distribution $p$ has a finite covariance, a version of Theorem 2 holds for some $\epsilon_{n,\delta}(p)$ that approaches the high-dimensional sub-Gaussian error as $n \to \infty$, up to constants. Further, for distributions without a finite covariance matrix, it is open to fully characterize the instance-by-instance error rates in high dimensions, and finding an analog of $\epsilon_{n,\delta}(p)$ that characterizes both the upper and lower bounds.

## 1.3  Related Work

The mean estimation problem has been extensively studied, even in one dimension. In the classic setting where the underlying distribution is assumed to have finite but unknown variance, the median-of-means algorithm, independently discovered by different authors [e.g., JVV86; AMS99; NY83], was the first to achieve sub-Gaussian rate to within a constant factor in the high probability regime. The seminal work of Catoni [Cat12] reinvigorated the study of mean estimation, by proposing the first estimator which attains sub-Gaussian rate tight to within a $1 + o(1)$ factor, but his estimator requires a-priori knowledge of the variance, or a bounded $4^{\text{th}}$ moment assumption that allows accurate estimation of the variance. This work further showed that the sub-Gaussian rate is a lower bound on the optimal estimation error. Subsequent work by Devroye et al. [DLLO16] proposed a different estimator, also attaining $1 + o(1)$-tight sub-Gaussian rate under the bounded $4^{\text{th}}$ moment assumption, which has additional structural properties. Recent work by Lee and Valiant [LV22] constructs an estimator achieving sub-Gaussian rate to within a $1 + o(1)$ factor for any distribution with finite but unknown variance, absent any knowledge assumption or bounded $4^{\text{th}}$ moment assumption.

In the more extreme setting where the underlying distribution may have infinite variance, but is guaranteed to have finite (but unknown) $1 + \alpha^{\text{th}}$ moment for some $\alpha \in (0, 1)$, Bubeck et al. [BCL13] proved an upper bound on the error achieved the median-of-means estimator, and Devroye et al. [DLLO16] showed a matching lower bound up to a constant factor.

See the in-depth survey by Lugosi and Mendelson [LM19] on sub-Gaussian mean estimation and regression results prior to 2019.

"Beyond worst-case" analysis is a theme of much recent work and attention in the computer science literature. See [Rou21] for examples of "beyond worst-case" analyses in various contexts in computer science and statistics including, for example, combinatorial algorithmic problems, auction design, and hypothesis testing. Both of the above tightness results in mean estimation are in the worst-case, and in this work, we present (to our knowledge) the first "beyond worst-case analysis" results for the mean estimation problem. We emphasize also that our results are applicable even to distributions with a finite mean, but without any finite $1 + \alpha^{\text{th}}$ moment for any $\alpha > 0$.

The notion of admissibility in statistics and the analogous concept of Pareto efficiency serve as a main motivation for our definition of neighborhood optimality, introduced in Section 3. They are well-studied notions in their respective fields, to the extent that they are standard topics in undergraduate courses. See for example the textbooks by Keener [Kee10] and Starr [Sta97] for expositions.

The other main definitional motivation is the notion of instance optimality, which falls under the umbrella of "beyond worst-case analysis" in the computer science literature. We highlight some of the uses of instance optimality in statistical contexts. Valiant and Valiant [VV17] gave the first instance optimal algorithm for the identity testing problem (in total variation distance) for discrete distributions. In later work [see VV16], the same authors showed how to instance-optimally learn discrete distributions (in total variation distance).

A different line of work studies "instance optimality" in the context of mean estimation with differential privacy (DP) [e.g., AD20b; AD20a; HLY21]. Specifically, these works address the problem of differentially-privately estimating the mean of a data set, where the data set itself (or equivalently, the uniform distribution over the data set) is the instance. In the DP setting, instance optimality is also not satisfiable by any estimator for the same reason as the non-DP setting, since the hardcoded estimator is always differentially private. They instead use a *local minimax* (or locally worst-case) notion of optimality (although in these works this notion is sometimes also called "instance optimality"; we take care to distinguish the two definition styles in this paper), where the particular locality/neighborhood

structure they use is restricted to data sets that have Hamming distance 1 from the instance, since this is a key component in the definition of DP. The local minimax notion of optimality is closely related to our notion of neighborhood optimality. We compare and contrast the two notions in Appendix A.3, and explain why our definition is stronger and more appropriate for our context.

## 2 Proof of main results

This section gives the construction and proof of our main result, Theorem 2.

Our construction of $q$ from $p$ will have 2 cases, depending on which of the 2 terms in the definition of the error function $\epsilon_{n,\delta}$ dominates. Recall that the error $\epsilon_{n,\delta}(p)$ is the sum of two terms involving the "trimmed" distribution $p_n^*$: (ignoring constants) $|\mu_p - \mu_{p_n^*}|$ and $\sigma_{p_n^*}\sqrt{\log\frac{1}{\delta}/n}$; intuitively, the first term measures to what degree $p$ has an "asymmetric tail", and the second term measures the variance of $p$ in its central region.

First, consider the case when the first term is larger, namely, $|\mu_p - \mu_{p_n^*}| > \sigma_{p_n^*}\sqrt{4.5\log\frac{1}{\delta}/n}$. Our goal in constructing $q$ is to maximize $|\mu_p - \mu_q|$ subject to $q$ being indistinguishable from $p$. Given that 1) the mean of $p_n^*$ is already far from the mean of $p$ by assumption in the case analysis, and 2) $p$ and $p_n^*$ are by construction hard to distinguish since only a small amount of probability mass was trimmed from $p$ to make $p_n^*$, we simply need to construct $q$ as a carefully chosen convex combination of $p$ and $p_n^*$, and show that this $q$ indeed satisfies all the properties in the definition of $N_{n,\delta}(p)$.

Next, for the remaining case when the variance term $\sigma_{p_n^*}\sqrt{4.5\log\frac{1}{\delta}/n}$ is larger than the remaining term $|\mu_p - \mu_{p_n^*}|$ in $\epsilon_{n,\delta}(p)$, we now want to construct $q$ such that the mean shift $|\mu_p - \mu_q|$ is large compared to the variance term $\sigma_{p_n^*}\sqrt{4.5\log\frac{1}{\delta}/n}$, while ensuring that $q$ is indistinguishable from $p$. To achieve this, we create $q$ that is a "skewed" version of $p$, scaling the probability density by a linear function $1 + ax$, where larger $a$ means more mean shift between $q$ and $p$, but also means that it is easier to distinguish $q$ from $p$. Technically, we truncate the linear scaling factor $1 + ax$ to lie between 0 and 2, so that $q$ will satisfy Condition 3 of the requirements for $q$ presented in our main theorem, Theorem 2; the probability mass might not be 1 after this skewing, so we might need to normalize; also, we point out that for the purposes of Theorem 2 we do not care whether the mean of $q$ is shifted to the *left* or *right* of $p$ (corresponding to choosing a positive or negative parameter $a$), and the construction makes use of this choice.

**Definition 4 (Construction of indistinguishable pair)** *Given a distribution $p$, we construct a distribution $q$ in a shift-and-scale invariant manner as follows.*

**Case 1:** $|\mu_p - \mu_{p_n^*}| > \sigma_{p_n^*}\sqrt{4.5\frac{\log\frac{1}{\delta}}{n}}$. *Define $\lambda = \frac{3}{4}$ and construct $q$ to be the weighted average $q = \lambda p + (1 - \lambda)p_n^*$.*

**Case 2:** $|\mu_p - \mu_{p_n^*}| \leq \sigma_{p_n^*}\sqrt{4.5\frac{\log\frac{1}{\delta}}{n}}$. *Without loss of generality, assume that $\mu_p = 0$. Let parameter $a$ be the solution in the interval $\left(0, \frac{1}{\sigma_{p_n^*}}\sqrt{\frac{\log\frac{1}{\delta}}{n}}\right]$ to the below equation characterizing the mean shift, as guaranteed by Lemma 23 in Appendix B:*

$$\int_{-\infty}^{-\frac{1}{a}}(-x)\,\mathrm{d}p + a\cdot\int_{-\frac{1}{a}}^{\frac{1}{a}}x^2\,\mathrm{d}p + \int_{\frac{1}{a}}^{\infty}x\,\mathrm{d}p = \frac{1}{8}\sigma_{p_n^*}\sqrt{\frac{\log\frac{1}{\delta}}{n}}$$

*We construct two non-unit measures $q^+, q^-$, defined as scaled versions of $p$, as $\frac{\mathrm{d}q^\pm}{\mathrm{d}p}(x) = 1 + \min(1, \max(-1, \pm ax))$. By symmetry the masses of $q^+$ and $q^-$ sum to 2; thus one of $q^+, q^-$ has mass at least 1. Construct $q$ by choosing that one of $q^+, q^-$, and downscaling it by a factor $b \in [\frac{1}{2}, 1]$ such that the total probability mass is indeed 1.* ◁

In the rest of the subsections, we will prove all the properties of the construction needed by Theorem 2. We observe that the non-unit measure $q^-$ is "symmetric" to $q^+$, in the sense that its first moment shift

from $p$ is identical but of the opposite sign. Therefore, most of the analysis below will assume the $q^+$ case without loss of generality.

## 2.1 Checking that $\frac{\mathrm{d}q}{\mathrm{d}p} \leq 2$

It is straightforward to check that $\frac{\mathrm{d}q}{\mathrm{d}p} \leq 2$ by construction, for both cases of Definition 4.

**Lemma 5** *Suppose there is a sufficiently small absolute constant that upper bounds $\frac{\log \frac{1}{\delta}}{n}$. Given a distribution $p$, if we construct $q$ as in Case 1 (the large $|\mu_{p_n^*} - \mu_p|$ case) in Definition 4, then $\frac{\mathrm{d}q}{\mathrm{d}p} \leq 2$.*

**Proof.** Let the support of $p_n^*$ be $[x_{\mathrm{left}}, x_{\mathrm{right}}]$. At $x \notin [x_{\mathrm{left}}, x_{\mathrm{right}}]$, we have $\frac{\mathrm{d}q}{\mathrm{d}p} = \frac{\lambda \, \mathrm{d}p}{\mathrm{d}p} = \frac{3/4 \, \mathrm{d}p}{\mathrm{d}p} \leq 2$. Otherwise, at $x \in [x_{\mathrm{left}}, x_{\mathrm{right}}]$, we have

$$\frac{\mathrm{d}q}{\mathrm{d}p} = \frac{\lambda \, \mathrm{d}p + (1 - \lambda) \, \mathrm{d}p_n^*}{\mathrm{d}p} = \frac{\mathrm{d}}{\mathrm{d}p} \left( \lambda p + \frac{1 - \lambda}{1 - \frac{0.45 \log \frac{1}{\delta}}{n}} p \right) \quad \text{(by the definition of } p_n^*) \leq 2$$

where the last line follows from the fact that $\lambda$ is a constant in $(0, 1)$ and $\frac{\log \frac{1}{\delta}}{n}$ is assumed to be bounded by some sufficiently small absolute constant. $\qquad \square$

**Lemma 6** *Suppose there is a sufficiently small absolute constant that upper bounds $\frac{\log \frac{1}{\delta}}{n}$. Given a distribution $p$, if we construct $q$ as in Case 2 (the small $|\mu_{p^*} - \mu_p|$ case) in Definition 4, then $\frac{\mathrm{d}q}{\mathrm{d}p} \leq 2$.*

**Proof.** For $q^+$, we have $\frac{\mathrm{d}q^+}{\mathrm{d}p}(x) = 1 + \min(1, \max(-1, a^+ x))$ for some value of $a^+$, and the right hand side is always between 0 and 2; proven similarly, we have $0 \leq \frac{\mathrm{d}q^-}{\mathrm{d}p} \leq 2$. Noting that $q$ is constructed by scaling down one of $q^-$ and $q^+$, we also have $\frac{\mathrm{d}q}{\mathrm{d}p} \leq 2$. $\qquad \square$

## 2.2 Bounding the squared Hellinger distance

We consider each case of the construction in Definition 4 separately. We first bound the Hellinger distance between $p$ and the $q$ constructed in Case 1 of Definition 4 (when $|\mu_p - \mu_{p_n^*}|$ is large).

**Lemma 7** *Suppose there is a sufficiently small constant that upper bounds both $\frac{\log \frac{1}{\delta}}{n}$ and $\delta$. Given a distribution $p$, if we construct $q$ as in Case 1 (the large $|\mu_{p_n^*} - \mu_p|$ case) in Definition 4, then $\log(1 - d_{\mathrm{H}}^2(p, q)) \geq \frac{1}{2n} \log 4\delta$.*

Since Case 1 of the construction of $q$ in Definition 4 linearly interpolates between $p$ and—a very slightly trimmed version of $p$, namely—$p_n^*$, the resulting distribution $q$ remains close to $p$; the calculation is in Appendix B.1. The next lemma bounds the squared Hellinger distance of $p$ and $q$ in Case 2 of Definition 4 (when $|\mu_p - \mu_{p_n^*}|$ is small).

**Lemma 8** *Suppose there is a sufficiently small constant that upper bounds both $\frac{\log \frac{1}{\delta}}{n}$ and $\delta$. Given a distribution $p$, if we construct $q$ as in Case 2 (the small $|\mu_{p^*} - \mu_p|$ case) in Definition 4, then $\log(1 - d_{\mathrm{H}}^2(p, q)) \geq \frac{1}{2n} \log 4\delta$.*

The proof (see Appendix B.1) uses a technical lemma to relate $d_H(p, q)$ to $d_H(p, q^+)$ or $d_H(p, q^-)$, and then uses a linearization of the definition of Hellinger distance to relate it to the mean shift between $p$ and $q^+$ or $q^-$, which is bounded by Definition 4.

## 2.3 Lower bounding $|\mu_q - \mu_p|$

We show the lower bound separately for the two cases in the construction of $q$ in Definition 4. The small $|\mu_{p_n^*} - \mu_p|$ case is a direct corollary of Lemma 23, which bounds the mean shift in this case.

**Lemma 9** *Suppose there is a sufficiently small constant that upper bounds both $\frac{\log \frac{1}{\delta}}{n}$ and $\delta$. Given a distribution $p$, if we construct $q$ as in Case 1 (the large $|\mu_{p_n^*} - \mu_p|$ case) in Definition 4, then $|\mu_q - \mu_p| \geq \frac{1}{8}\epsilon_{n,\delta}(p)$.*

**Proof.** Without loss of generality, assume that $\mu_p = 0$. Recall that $q = \lambda p + (1 - \lambda)p_n^*$ and $\lambda = \frac{3}{4}$ from Case 1 of Definition 4. Thus, $|\mu_q - \mu_p| = \frac{1}{4}|\mu_{p_n^*} - \mu_p|$. Furthermore, we have $\epsilon_{n,\delta}(p) = |\mu_{p_n^*} - \mu_p| + \sigma_{p_n^*}\sqrt{4.5\frac{\log \frac{1}{\delta}}{n}}$ from the definition of $\epsilon_{n,\delta}(p)$ and $|\mu_{p_n^*} - \mu_p| > \sigma_{p_n^*}\sqrt{4.5\frac{\log \frac{1}{\delta}}{n}}$ from the lemma assumption and Case 1 of Definition 4, which then imply that $\epsilon_{n,\delta}(p) \leq 2|\mu_{p_n^*} - \mu_p|$. Combining the two inequalities yields $|\mu_q - \mu_p| \geq \frac{1}{8}\epsilon_{n,\delta}(p)$ as desired. $\qquad\square$

**Lemma 10** *Suppose there is a sufficiently small constant that upper bounds both $\frac{\log \frac{1}{\delta}}{n}$ and $\delta$. Given a distribution $p$, if we construct $q$ as in Case 2 (the small $|\mu_{p_n^*} - \mu_p|$ case) in Definition 4, then $|\mu_q - \mu_p| \geq \frac{1}{32}\epsilon_{n,\delta}(p)$.*

**Proof.** Without loss of generality, let $q^+$ be the non-unit measure used to construct $q$, that is $q = bq^+$ for some $b \in [\frac{1}{2}, 1]$. By the bound on $b$ as well as the construction of $q^+$ in Definition 4, we have $|\mu_q - \mu_p| = b|\int x\,dq^+ - \int x\,dp| \geq \frac{1}{16}\sigma_{p_n^*}\sqrt{\frac{\log \frac{1}{\delta}}{n}}$. Additionally, we both have $\epsilon_{n,\delta}(p) = |\mu_{p_n^*} - \mu_p| + \sigma_{p_n^*}\sqrt{4.5\frac{\log \frac{1}{\delta}}{n}}$ from the definition of $\epsilon_{n,\delta}(p)$ and $|\mu_{p_n^*} - \mu_p| \leq \sigma_{p_n^*}\sqrt{4.5\frac{\log \frac{1}{\delta}}{n}}$ from the lemma assumption and Case 2 of Definition 4, both of which imply $\epsilon_{n,\delta}(p) \leq 2\sigma_{p_n^*}\sqrt{4.5\frac{\log \frac{1}{\delta}}{n}}$. Combining the two inequalities above, we have that $|\mu_q - \mu_p| \geq \frac{1}{32}\epsilon_{n,\delta}(p)$. $\qquad\square$

## 3 Neighborhood Optimality: A New Definition Framework

Our main result is a specific and technical indistinguishability result. This section aims to clarify, through a new definition framework, the optimality notion that our technical result implies.

### 3.1 Neighborhood Optimality

Usual notions of "beyond worst-case optimality" include "instance optimality" which is unattainably strong, and "admissibility"/"Pareto efficiency" from the statistics and economics literature, which is too weak. In particular, the latter notion is too weak in the sense that a trivial estimator that always outputs the same hardcoded mean estimate is actually admissible, despite being algorithmically "vacuous". We define and explain these notions formally in Appendix D.

Given that neither of the usual definitions are suitable for mean estimation, in this work we give a new optimality definition, which we call *neighborhood optimality*. We state its definition in this section, and explore its basic properties and intuition in Appendix A, including how the definition smoothly interpolates between instance optimality and admissibility. Our definition is also related to the notion of local minimax optimality, which we compare with in Appendix A.3. The differences are subtle, yet, as we show in Appendix A.3, local minimax is *too weak* a notion and, when instantiated inappropriately, allows for absurd bounds to be proven. We thus advocate for this new optimality definition, which correctly rejects such absurd bounds. As an application of our new framework, we prove in Section 4 that the median-of-means estimator is neighborhood optimal up to constant factors. It is an open question to find a neighborhood optimal estimator *without* the constant factor slackness.

Let $\mathcal{P}_1$ be the entire set of all distributions with a finite first moment over $\mathbb{R}$. We say that $N$ is a neighborhood function (defined over $\mathcal{P}_1$) if $N$ maps a distribution $p \in \mathcal{P}_1$ to a set of distributions $N(p) \subseteq \mathcal{P}_1$. For the purposes of the rest of the definitions, it will not matter whether $p \in N(p)$. Similarly, an error function $\epsilon$ maps distributions to non-negative numbers, like $\epsilon_{n,\delta}$ in our main result, Theorem 2. In the later definitions, we use the notations $N_{n,\delta}$ and $\epsilon_{n,\delta}$ to denote their dependence on the sample complexity $n$ and failure probability $\delta$.

Given these two notions, we can now define *neighborhood Pareto bounds with respect to $N_{n,\delta}$*, which imposes admissibility structure within the local neighborhood $N_{n,\delta}(p)$ of every distribution $p \in \mathcal{P}_1$.

**Definition 11 (Neighborhood Pareto bounds with respect to $N_{n,\delta}$)** *Let $n$ be the number of samples and $\delta$ be the failure probability. Given a neighborhood function $N_{n,\delta} : \mathcal{P}_1 \to 2^{\mathcal{P}_1}$, we say that the error function $\epsilon_{n,\delta}(p) : \mathcal{P}_1 \to \mathbb{R}_0^+$ is a neighborhood Pareto bound for $\mathcal{P}_1$ with respect to $N_{n,\delta}$ if for all distributions $p \in \mathcal{P}_1$, no estimator $\hat{\mu}$ taking $n$ i.i.d. samples can simultaneously achieve the following two conditions:*

- *For all $q \in N_{n,\delta}(p)$, with probability $1 - \delta$ over the $n$ i.i.d. samples from $q$, $|\hat{\mu} - \mu_q| \leq \epsilon_{n,\delta}(q)$.*

- *With probability $1 - \delta$ over the $n$ i.i.d. samples from $p$, $|\hat{\mu} - \mu_p| < \epsilon_{n,\delta}(p)$.*

Note the strict inequality in the second bullet: namely, it is impossible to "beat" the error function over an entire neighborhood, where "beating" is defined as attaining the error function over the neighborhood, and performing strictly better than the error function for $p$. The above two bullet points essentially capture admissibility within the local neighborhood $N_{n,\delta}(p) \cup \{p\}$—compare with Definition 28—and the definition requires admissibility within every such local neighborhood, over every possible $p$.

The neighborhood $N_{n,\delta}(p)$ in a neighborhood Pareto bound can be interpreted as the set of distributions "near $p$" which, if an estimator performs well on distribution $p$, then we should reasonably expect or want it to perform well also on all the distributions in the local neighborhood $N_{n,\delta}(p)$.

Using the notion of neighborhood Pareto bounds, we can now define $\kappa$-neighborhood optimal estimators, which are estimators whose performances are matched by neighborhood Pareto bounds.

**Definition 12 ($(\kappa, \tau)$-Neighborhood optimal estimators)** *Let $\kappa > 1$ be a multiplicative loss factor in estimation error, and $\tau > 1$ be a multiplicative loss factor in sample complexity. Given the parameters $\kappa, \tau > 1$, sample complexity $n$, failure probability $\delta$ and neighborhood function $N_{n,\delta}$, a mean estimator $\hat{\mu}$ is $(\kappa, \tau)$-neighborhood optimal with respect to $N_{n,\delta}$ if there exists an error function $\epsilon_{n,\delta}(p)$ such that $\min(\epsilon_{n/\tau,\delta}(p), \epsilon_{n,\delta}(p))$ is a neighborhood Pareto bound[2], and $\hat{\mu}$ gives estimation error at most $\kappa \cdot \epsilon_{n,\delta}(p)$ with probability at least $1 - \delta$ when taking $n$ i.i.d. samples from any distribution $p \in \mathcal{P}_1$.*

As a basic example and sanity check, in Appendix E, we show that any trivial estimator that outputs a hardcoded mean estimate cannot be $\kappa$-neighborhood optimal with respect to our chosen neighborhood function (Definition 15 in Section 4) for any $\kappa$.

### 3.2 Indistinguishability implies a neighborhood Pareto bound

Even though it might not look obvious how we can prove a neighborhood Pareto bound from its definition, we show that our main indistinguishability result essentially implies such a bound. The proof essentially follows the straightforward estimation-to-testing reduction intuition, and we give it formally in Appendix A.2.

**Proposition 13 ("Local" indistinguishability bounds imply neighborhood Pareto bounds)** *The error function $\epsilon_{n,\delta}$ is a neighborhood Pareto bound with respect to the neighborhood function $N_{n,\delta}$ if for every distribution $p \in \mathcal{P}_1$, there exists a distribution $q \in N_{n,\delta}(p)$, with $q \neq p$, such that $|\mu_p - \mu_q| \geq \epsilon_{n,\delta}(p) + \epsilon_{n,\delta}(q)$ and it is information-theoretically impossible to distinguish $p$ and $q$ with probability $1 - \delta$ using $n$ samples.*

## 4 Median-of-Means is Neighborhood Optimal

To apply our new definitional framework, we choose a reasonable neighborhood function $N_{n,\delta}$ and show that the median-of-means algorithm is neighborhood optimal with respect to this choice.

In Appendix C, we give the following (straightforward) re-analysis of median-of-means, which will form the upper bound part for neighborhood optimality.

---

[2]While it is intuitive to expect that an error function decreases in $n$, it might not be true in general. Indeed, the definition of $\epsilon_{n,\delta}(p)$ we use in the main result is not necessarily monotonic. This is why we use a $\min$ in the neighborhood Pareto bound requirement.

**Proposition 14** *Consider a distribution $p$ with mean $\mu_p$, a sample size $n$, and a median-of-means group count $4.5\log\frac{1}{\delta}$. Let $p_n^*$ be the $\frac{0.45}{n}\log\frac{1}{\delta}$-trimmed distribution from Definition 1, and $\mu_{p_n^*}$ and $\sigma_{p_n^*}$ be the mean and standard deviation of $p_n^*$ respectively. Then, the median-of-means estimator has error $|\mu_p - \mu_{p_n^*}| + 3\sigma_{p_n^*}\sqrt{\frac{4.5\log\frac{1}{\delta}}{n}}$ except with probability at most $\delta$.*

We can now discuss the neighborhood choice for the corresponding neighborhood Pareto bound. Recall that, intuitively, neighborhood optimality is asking "how well can our algorithm do on $p$ given that we also want our algorithm to do similarly well on a neighborhood of $p$"; and thus, the smaller we choose the neighborhood, the stronger the resulting theorem. We thus define the neighborhood of $p$ to consist of distributions that are similar or similarly nice to $p$ in 4 different ways:

**Definition 15 (Choice of neighborhood function $N_{n,\delta}$ in Theorem 16)** *Define $N_{n,\delta}(p)$ to be the set of distributions $q \in \mathcal{P}_1$ such that*

1. *$\epsilon_{n/3,\delta}(q) \leq 100\epsilon_{n,\delta}(p)$*

2. *$\log(1 - d_{\mathrm{H}}^2(p,q)) \geq \frac{1}{2n}\log 4\delta$*

3. *$|\mu_q - \mu_p| \leq \epsilon_{n,\delta}(p)$*

4. *For all $x \in \mathbb{R}$, $\frac{\mathrm{d}q}{\mathrm{d}p}(x) \leq 2$.*

As a basic sanity check, we show in Appendix E that a trivial, hardcoded estimator cannot be neighborhood optimal—this is mostly a consequence of Property 3 above. See Appendix E for a formal statement and proof. We then show:

**Theorem 16** *Let $n$ be the number of samples and $\delta$ be the failure probability. Assume that there is a sufficiently small constant which upper bounds both $\frac{\log\frac{1}{\delta}}{n}$ and $\delta$.*

*Consider the neighborhood function $N_{n,\delta}$ of Definition 15. Recall the error function defined in Definition 1 as $\epsilon_{n,\delta}(p) = |\mu_p - \mu_{p_n^*}| + \sigma_{p_n^*}\sqrt{\frac{4.5\log\frac{1}{\delta}}{n}}$. Then, for some sufficiently large constant $\kappa$, the error function $\frac{1}{\kappa}\min(\epsilon_{n/3,\delta}, \epsilon_{n,\delta})$ is a neighborhood Pareto bound with respect to $N_{n,\delta}$.*

*Combined with Proposition 14 stating that the median-of-means estimator has error function $O(\epsilon_{n,\delta})$, this implies the median-of-means estimator is $(\kappa, 3)$-neighborhood optimal with respect to $N_{n,\delta}$.*

The main component of the proof is our construction of $q$, and the accompanying analysis of Theorem 2 showing that $q$ is well behaved in several senses. We specifically show Lemma 17, a slight extension of Theorem 2:

**Lemma 17** *Let $n$ be the sample complexity and $\delta$ be the failure probability, and recall the definition of $\epsilon_{n,\delta}$ from Definition 1. Assume that there is a sufficiently small constant which upper bounds both $\frac{\log\frac{1}{\delta}}{n}$ and $\delta$. Then for any distribution $p$, there exists a distribution $q \neq p$ such that the mean of $q$ is $\frac{1}{32}\epsilon_{n,\delta}(p)$ different from the mean of $p$, and $\log(1 - d_{\mathrm{H}}^2(p,q)) \geq \frac{1}{2n}\log 4\delta$, and $q \in N_{n,\delta}(p)$.*

See Appendix F for the proof of Lemma 17. We will now use Lemma 17 to prove Theorem 16.

**Proof of Theorem 16.** By Proposition 13, it suffices to show that, for every distribution $p \in \mathcal{P}_1$, there exists a distribution $q \in N_{n,\delta}(p)$ with $q \neq p$ such that $|\mu_p - \mu_q| \geq \frac{1}{\kappa}(\min(\epsilon_{n/3,\delta}, \epsilon_{n,\delta})(p) + \min(\epsilon_{n/3,\delta}, \epsilon_{n,\delta})(q))$ for some large constant $\kappa$, and no tester can distinguish $p$ and $q$ with probability $1 - \delta$ using $n$ samples.

Given a distribution $p \in \mathcal{P}_1$, consider the distribution $q \in N_{n,\delta}(p)$ guaranteed by Lemma 17. Since $q$ satisfies $\log(1 - d_{\mathrm{H}}^2(p,q)) \geq \frac{1}{2n}\log 4\delta$, by Fact 1 we know that $p$ and $q$ are indistinguishable with probability $1 - \delta$ using $n$ samples.

It remains to check that $|\mu_p - \mu_q| \geq \frac{1}{\kappa}(\min(\epsilon_{n/3,\delta}, \epsilon_{n,\delta})(p) + \min(\epsilon_{n/3,\delta}, \epsilon_{n,\delta})(q))$ for some sufficiently large constant $\kappa$. By Lemma 17, we have

$$|\mu_p - \mu_q| \geq \Omega(\epsilon_{n,\delta}(p)) \geq \Omega(\epsilon_{n,\delta}(p) + \epsilon_{n,\delta}(p)) \geq \Omega(\epsilon_{n/3,\delta}(q) + \epsilon_{n,\delta}(p)) \quad \text{by Lemma 17}$$
$$\geq \Omega(\min(\epsilon_{n/3,\delta}, \epsilon_{n,\delta})(p) + \min(\epsilon_{n/3,\delta}, \epsilon_{n,\delta})(q))$$

which completes the proof of Theorem 16. $\qquad\square$

## Acknowledgements

We thank the anonymous reviewers for insightful comments and suggestions on this work. Jasper C.H. Lee is supported in part by the generous funding of a Croucher Fellowship for Postdoctoral Research, NSF award DMS-2023239, NSF Medium Award CCF-2107079 and NSF AiTF Award CCF-2006206. Maoyuan Song is supported in part by NSF award CCF-1910411, NSF award CCF-2228814, and NSF award CCF-2127806. Paul Valiant is supported by NSF award CCF-2127806.

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

# A    Interpreting Neighborhood Optimality

In this section, we derive basic properties and intuitions about the definitions of neighborhood Pareto bounds and neighborhood optimality.

In particular, we show that neighborhood optimality is a notion that smoothly interpolates between instance optimality and admissibility, depending on what neighborhood function is used to instantiate the definition. We also show a sufficient condition for proving that an error function $\epsilon$ is a neighborhood Pareto bound (Proposition 13). Lastly, we discuss and compare neighborhood optimality and the notion of local minimax optimality that has appeared in the literature.

## A.1    As an interpolation between instance optimality and admissibility

Recall that Definitions 11 and 12 both require specifying the neighborhood function. Our first observation is the *monotonicity* in the definition of neighborhood Pareto bounds, in the sense of the following straightforward proposition which we state without proof.

**Proposition 18** *Suppose the neighborhood functions $N_{n,\delta}$ and $N'_{n,\delta}$ are such that $N_{n,\delta}(p) \subseteq N'_{n,\delta}(p)$ for all $p$. Then, if a given error function $\epsilon_{n,\delta}$ is a neighborhood Pareto bound with respect to $N_{n,\delta}$, then $\epsilon_{n,\delta}$ is also a neighborhood Pareto bound with respect to $N'_{n,\delta}$.*

We can extend this monotonicity observation from neighborhood Pareto bounds to neighborhood optimal estimators.

**Proposition 19** *Suppose the neighborhood functions $N_{n,\delta}$ and $N'_{n,\delta}$ are such that $N_{n,\delta}(p) \subseteq N'_{n,\delta}(p)$ for all $p$. Then, if $\hat{\mu}$ is a $\kappa$-neighborhood optimal estimator with respect to $N_{n,\delta}$, then $\hat{\mu}$ is also $\kappa$-neighborhood optimal with respect to $N'_{n,\delta}$.*

To understand how the set of neighborhood optimal estimators vary as we change the neighborhood function, we examine the two extreme examples as special cases, where $N_{n,\delta} \equiv \emptyset$ and $N_{n,\delta} \equiv \mathcal{P}_1$.

Let us first consider the case when the neighborhoods are all empty. In this case, Definition 11 simplifies to requiring a neighborhood Pareto bound $\epsilon_{n,\delta}$ to be such that, for every distribution $p \in \mathcal{P}_1$, no estimator $\hat{\mu}$ (specialized to $p$) can have error strictly less than $\epsilon_{n,\delta}(p)$ with probability $1 - \delta$ over $n$ samples from $p$. It is straightforward to check that any estimator that is $\kappa$-neighborhood optimal with respect to empty neighborhoods is equivalent to being *instance optimal* (Definition 27 in Appendix D) up to a $\kappa$ factor, which as explained before, is impossible to achieve. The only possible neighborhood Pareto bound with respect to empty neighborhoods is $\epsilon_{n,\delta} \equiv 0$, since every distribution has a trivial estimator that outputs its mean hardcoded.

At the other extreme, consider the case where all the neighborhoods contain all the distributions in $\mathcal{P}_1$. Here, Definition 11 simplifies (after taking a contrapositive) to requiring a neighborhood Pareto bound $\epsilon$ to be such that, for every distribution $p \in \mathcal{P}_1$, if an estimator has error at most $\epsilon_{n,\delta}(q)$ with probability $1 - \delta$ for all distributions $q \neq p \in \mathcal{P}_1$, then it must have error at least $\epsilon_{n,\delta}(p)$ with probability $1 - \delta$ for distribution $p$. It is again straightforward to check that, by definition, an estimator is 1-neighborhood optimal with respect to the constant-$\mathcal{P}_1$ neighborhood function if and only if the estimator is admissible in $\mathcal{P}_1$ (Definition 28 in Appendix D). As explained before, admissibility is a somewhat weak notion on estimators, and includes trivial hardcoded estimators.

Combining these two extreme cases and the monotonicity propositions, we have shown that neighborhood optimality is a notion which interpolates between instance optimality and admissibility. More technically, neighborhood optimality can be viewed as a homomorphism mapping the partial ordering of neighborhood functions (the ordering induced by set inclusion) to the partial ordering of sets of estimators (also ordered by set inclusion).

We remark that, while it may be tempting to view neighborhood Pareto bounds as *lower bounds*, and neighborhood optimality as exhibiting an estimator whose error function matches a lower bound up to a constant factor, such an interpretation is actually not valid. The reason is that, depending on the choice of the neighborhood function, given an estimator with error function $\epsilon_{n,\delta}^{\text{upper}}$ and a neighborhood Pareto bound $\epsilon_{n,\delta}^{\text{pareto}}$, it can be the case that on some distributions $p \in \mathcal{P}_1$, we have $\epsilon_{n,\delta}^{\text{pareto}}(p) > \epsilon_{n,\delta}^{\text{upper}}(p)$; namely, these Pareto bounds should *not* be viewed as lower bounds. A simple

example is the previous case where the neighborhood function is $\mathcal{P}_1$, in which case it is straightforward to check that for any (hardcoded) *constant* $\hat{\mu} \in \mathbb{R}$, the error function $\epsilon_{n,\delta}^{\text{pareto}}(p) = |\hat{\mu} - \mu_p|$ is a neighborhood Pareto bound. For any distribution $p$ whose mean is extremely far from the constant $\hat{\mu}$, the median-of-means algorithm will have accuracy better than $|\hat{\mu} - \mu_p|$. Because of such examples, we take care *not* to refer to our bounds as lower bounds, but instead call them Pareto bounds in this paper.

## A.2 Indistinguishability implies a neighborhood Pareto bound

We restate Proposition 13 from Section 3.2 here for clarity, and give the proof of it below.

**Proposition 13 ("Local" indistinguishability bounds imply neighborhood Pareto bounds)** *The error function $\epsilon_{n,\delta}$ is a neighborhood Pareto bound with respect to the neighborhood function $N_{n,\delta}$ if for every distribution $p \in \mathcal{P}_1$, there exists a distribution $q \in N_{n,\delta}(p)$, with $q \neq p$, such that $|\mu_p - \mu_q| \geq \epsilon_{n,\delta}(p) + \epsilon_{n,\delta}(q)$ and it is information-theoretically impossible to distinguish $p$ and $q$ with probability $1 - \delta$ using $n$ samples.*

**Proof.** Suppose the "if" condition in the proposition is true, yet, for the sake of contradiction, that $\epsilon_{n,\delta}$ is not a neighborhood Pareto bound. Then, there exists a distribution $p \in \mathcal{P}_1$ and an estimator $\hat{\mu}$ taking $n$ i.i.d. samples such that

- For all distributions $q \in N_{n,\delta}(p)$, with probability $1 - \delta$ over the $n$ i.i.d. samples from $q$, $|\hat{\mu} - \mu_q| \leq \epsilon_{n,\delta}(q)$.
- With probability $1 - \delta$ over the $n$ i.i.d. samples from $p$, $|\hat{\mu} - \mu_p| < \epsilon_{n,\delta}(p)$.

By the proposition condition, there must exist some distribution $q \in N_{n,\delta}(p)$ such that $|\mu_p - \mu_q| \geq \epsilon_{n,\delta}(p) + \epsilon_{n,\delta}(q)$ and it is information-theoretically impossible to distinguish $p$ and $q$ with probability $1 - \delta$ using $n$ samples. However, we can construct the following distinguisher: compute a mean estimate $\hat{\mu}$, return $p$ if $\hat{\mu}$ is within $\epsilon_{n,\delta}(p)$ of $\mu_p$ and return $q$ otherwise. By our assumption on $\hat{\mu}$, this distinguisher will succeed with probability at least $1 - \delta$, thus contradicting the proposition statement. $\square$

We remark that the proof of Proposition 13 actually establishes a stronger result, that the error function $\epsilon_{n,\delta}$ is a neighborhood Pareto bound for the singleton neighborhood $N_{n,\delta}^*(p) = \{q(p)\}$ where $q(p)$ is the $q$ constructed from $p$ according to Proposition 13. Recall that the monotonicity property of Proposition 18 says that neighborhood Pareto bounds are stronger for smaller neighborhoods; thus this bound for singleton neighborhoods implies the corresponding bound for any larger neighboorhoods $N_{n,\delta} \supset N_{n,\delta}^*$.

## A.3 Comparing with local minimax optimality

Here, we compare our definition of neighborhood optimality with the notion of local minimax optimality from prior literature. At a high level, neighborhood optimality imposes admissibility within each local neighborhood, whereas local minimax optimality imposes minimax optimality within each local neighborhood. We argue that local minimax is *too* sensitive to the choice of neighborhood structure, although the two definitions are different in a rather subtle way and perhaps difficult to see at first glance. We concretely illustrate their differences via 1) a proposition showing that for practical purposes, local minimax bounds are a weaker notion than neighborhood Pareto bounds and 2) a simple example choice of an "inappropriate" neighborhood structure, such that an intuitively absurd local minimax bound holds for this neighborhood structure, but the same bound fails to satisfy the definition of a neighborhood Pareto bound. Together, these results show that neighborhood optimality is a stronger and more robust notion than local minimax optimality.

For ease of comparison, we first phrase the local minimax definition in the same form as our definition of neighborhood optimality.

**Definition 20 (Local minimax bounds with respect to $N_{n,\delta}$)** *Let $n$ be the number of samples and $\delta$ be the failure probability. Given a neighborhood function $N_{n,\delta} : \mathcal{P}_1 \to 2^{\mathcal{P}_1}$, we say that the error function $\epsilon_{n,\delta}(p) : \mathcal{P}_1 \to \mathbb{R}_0^+$ is a local minimax bound for $\mathcal{P}_1$ with respect to $N_{n,\delta}$ if for all*

distributions $p \in \mathcal{P}_1$, there is no *estimator $\hat{\mu}$ such that for all distributions $q \in N_{n,\delta}(p) \cup \{p\}$, when given $n$ i.i.d. samples from $q$, the estimator $\hat{\mu}$ achieves $|\hat{\mu} - \mu_q| < \epsilon_{n,\delta}(p)$ with probability $1 - \delta$.*

**Definition 21 ($\kappa$-locally minimax estimators)** *For a parameter $\kappa > 1$, sample complexity $n$, failure probability $\delta$ and neighborhood function $N_{n,\delta}$, a mean estimator $\hat{\mu}$ is $\kappa$-locally minimax with respect to $N_{n,\delta}$ if there exists an error function $\epsilon_{n,\delta}(p)$ such that $\epsilon_{n,\delta}(p)$ is a local minimax bound, and $\hat{\mu}$ gives estimation error at most $\kappa \cdot \epsilon_{n,\delta}(p)$ with probability at least $1 - \delta$ when taking $n$ i.i.d. samples from any distribution $p \in \mathcal{P}$.*

The notions of local minimax bounds and locally minimax estimators also have the same monotonicity properties as neighborhood Pareto bounds and neighborhood optimal estimators, analogous to Propositions 18 and 19. When the neighborhood function $N_{n,\delta}$ is constantly equal to the empty set, local minimax optimality again coincides with instance optimality. Furthermore, local minimax bounds can be proved using indistinguishability arguments, via a reduction analogous to Proposition 13 up to minor changes in parameters.

However, one key difference between the two styles of definition arises when we consider proving neighborhood Pareto bounds or local minimax bounds using such indistinguishability arguments. Recall from Section 3.2 that, by relying on Proposition 13, the main technical result of this paper is to show that for every distribution $p$, there exists a "neighbor" $q(p)$ whose mean is far from $p$, with $q(p)$ satisfying suitable structural properties (so that $q(p) \in N_{n,\delta}(p)$ in our eventual choice of $N_{n,\delta}$ in Definition 15), such that $p$ and $q$ cannot be distinguished with probability $1 - \delta$ using $n$ samples. As explained in Section 3.2, this proof strategy effectively proves a neighborhood Pareto bound over the singleton neighborhood function $N_{n,\delta}^*(p) = \{q(p)\}$. Thus, it is meaningful to compare neighborhood Pareto bounds and local minimax bounds when the neighborhoods are singletons. The following proposition shows that, up to a constant factor of 2 in the error, a neighborhood Pareto bound on singleton neighborhoods implies a potentially much larger local minimax bound.

**Proposition 22 (For singleton neighborhoods, local minimax bounds are weaker than neighborhood Pareto bounds)** *Let $n$ be the number of samples and $\delta$ be the failure probability. Consider a neighborhood function $N_{n,\delta}^*$ such that for all $p \in \mathcal{P}_1$, $N_{n,\delta}^*(p)$ is a singleton set containing a distribution $q(p) \neq p$. Suppose $\epsilon_{n,\delta}$ is a neighborhood Pareto bound with respect to $N_{n,\delta}^*$, and that $\epsilon_{n,\delta}(p) > 0$ for all $p \in \mathcal{P}_1$. Then, the function $\frac{1}{2}(\epsilon_{n,\delta}(p) + \epsilon_{n,\delta}(q(p)))$ is a local minimax bound with respect to $N_{n,\delta}^*$. Note that the above function is lower bounded by $\Omega(\max(\epsilon_{n,\delta}(p), \epsilon_{n,\delta}(q(p))))$ and can be much larger than $\epsilon_{n,\delta}(p)$.*

**Proof.** Suppose for the sake of contradiction that $\frac{1}{2}(\epsilon_{n,\delta}(p) + \epsilon_{n,\delta}(q(p)))$ is *not* a local minimax bound with respect to $N_{n,\delta}^*$ but $\epsilon_{n,\delta}$ is a neighborhood Pareto bound.

We observe that, since $\epsilon_{n,\delta}$ is a neighborhood Pareto bound, it must be the case that for every distribution $p \in \mathcal{P}_1$, we have $|\mu_p - \mu_q| \geq \epsilon_{n,\delta}(p) + \epsilon_{n,\delta}(q(p))$. Otherwise, for any distribution $p$ not satisfying the above, there is a trivial hardcoded estimator that outputs a number $\hat{\mu}$ such that $|\hat{\mu} - \mu_q| < \epsilon_{n,\delta}(q(p))$ and $|\hat{\mu} - \mu_p| < \epsilon_{n,\delta}(p)$.

Since $\frac{1}{2}(\epsilon_{n,\delta}(p) + \epsilon_{n,\delta}(q(p)))$ is not a local minimax bound, there exists some distribution $p$ and some estimator $\hat{\mu}$ such that 1) with probability at least $1 - \delta$ over $n$ samples from $p$, $|\hat{\mu} - \mu_p| < \frac{1}{2}(\epsilon_{n,\delta}(p) + \epsilon_{n,\delta}(q(p)))$ and 2) the same for $q$. However, we already know that $|\mu_p - \mu_q| \geq \epsilon_{n,\delta}(p) + \epsilon_{n,\delta}(q(p))$, which implies that we can use the mean estimator $\hat{\mu}$ to distinguish $p$ and $q$ with probability $1 - \delta$ over $n$ samples. Using this distinguisher, we construct a new estimator $\hat{\mu}'$ which outputs $\mu_p$ if the distinguisher thinks the distribution is $p$, and $\mu_q$ otherwise. This new estimator $\hat{\mu}'$ has 0 error with probability $1 - \delta$ over $n$ samples, which contradicts the assumption that $\epsilon_{n,\delta}$ is a neighborhood Pareto bound and that $\epsilon_{n,\delta}(p) > 0$. $\square$

The notion of local minimax bounds is therefore (potentially much) weaker than the notion of a neighborhood Pareto bound we introduce in this work. We now show that this *can* actually happen, if we do not choose the neighborhood structure carefully. We give a concrete example of a neighborhood structure in which an absurdly large local minimax bounds holds, but this bad bound is (rightfully) rejected by the definition of neighborhood Pareto bounds.

As a representative simple example, consider $p = \mathcal{N}(0, 1)$ and define its neighborhood as a singleton set $N(p) = \{q = \mathcal{N}(\eta, 1)\}$ where $\eta \ll 1$ is small enough that $p$ and $q$ are indistinguishable with

$n$ samples. Now define the neighborhood of $q$ to also be a singleton set $N(q) = \{q'\}$, where $q'$ is constructed by moving a tiny bit of mass of $q$ such that $\mu_{q'} = \eta + 10^6$, but $q$ and $q'$ are indistinguishable. Consider an absurd error function $\epsilon(p) = \eta/2$, $\epsilon(q) = 10^6/2$, which is far too large for $q$ since we expect $O(\sqrt{\log \frac{1}{\delta}/n}) \ll 1$ estimation error for $q$ (e.g. by using a standard sub-Gaussian mean estimator). Yet, $\epsilon$ is a local minimax bound (c.f. Definition 21) under the neighborhood function $N$, since, given two indistinguishable distributions $p$ and $q$, no estimator can get error less $|\mu_p - \mu_q|/2$. On the other hand, we can also check that Definition 12 *rejects* this absurd $\epsilon$ error function from being a neighborhood Pareto bound. To see this, consider the neighborhood of $p$, consisting only of $q$. Consider the hardcoded estimator $\hat\mu$ always outputting 0: $\hat\mu$ violates the condition of neighborhood Pareto bounds since its error for $p$ is $|\hat\mu - \mu_p| = 0 < \eta/2$ and for $q$ is $|\hat\mu - \mu_q| = \eta \ll 10^6/2$.

This example, together with Proposition 22, show that neighborhood optimality is a more robust notion than local minimax optimality when being applied to inappropriately chosen neighborhood structures. While we believe our paper uses an "appropriate" neighborhood structure, we still emphasize the importance of introducing definitions that are properly *resilient* to absurd instantiations. For this reason, we have chosen to present our results in this paper as a neighborhood Pareto bound and optimality.

## B  Remaining proofs for Theorem 2

In Definition 4, we claimed that there exists a parameter $a$ satisfying certain conditions in Case 2 of the construction. We formally show that this parameter exists in the following lemma.

**Lemma 23** *Let $n$ be the number of samples and and $\delta$ be the failure probability, and assume that $\frac{\log \frac{1}{\delta}}{n}$ is bounded by some sufficiently small absolute constant. Let $p$ be any distribution such that $|\mu_p - \mu_{p_n^*}| \le \sigma_{p_n^*} \sqrt{c \frac{\log \frac{1}{\delta}}{n}}$, and we assume that $\mu_p = 0$ without loss of generality. Then, the equation*

$$\int_{-\infty}^{-\frac{1}{a}} (-x)\,\mathrm{d}p + a \cdot \int_{-\frac{1}{a}}^{\frac{1}{a}} x^2\,\mathrm{d}p + \int_{\frac{1}{a}}^{\infty} x\,\mathrm{d}p = \frac{1}{8}\sigma_{p_n^*}\sqrt{\frac{\log\frac{1}{\delta}}{n}}$$

*always has solution $a \in \left(0, \frac{1}{\sigma_{p_n^*}}\sqrt{\frac{\log\frac{1}{\delta}}{n}}\right]$.*

**Proof.** We first point out that, for the (potentially) non-unit measure $q^+$ defined by $\frac{\mathrm{d}q^+}{\mathrm{d}p}(x) = 1 + \min(1, \max(-1, ax))$, the left hand side of the equation equals the difference between the first moments of $q^+$ and $p$. Namely,

$$\int_{-\infty}^{\infty} x(\mathrm{d}q^+ - \mathrm{d}p) = \int_{-\infty}^{\infty} x\min(1, \max(-1, ax))\,\mathrm{d}p = \int_{-\infty}^{-\frac{1}{a}}(-x)\,\mathrm{d}p + a\cdot\int_{-\frac{1}{a}}^{\frac{1}{a}} x^2\,\mathrm{d}p + \int_{\frac{1}{a}}^{\infty} x\,\mathrm{d}p$$

Our goal is to find $a \in \left(0, \frac{1}{\sigma_{p_n^*}}\sqrt{\frac{\log\frac{1}{\delta}}{n}}\right]$ such that this first moment shift equals $\frac{1}{8}\sigma_{p_n^*}\sqrt{\frac{\log\frac{1}{\delta}}{n}}$.

Because $\min(1, \max(-1, ax))$ is increasing in $a$ and continuous in $a$, the first moment shift is also increasing and continuous in $a$. When $a = 0$ then $q^+ = p$, and the shift clearly equals 0. Thus it suffices for us to show that the first moment shift is at least $\frac{1}{8}\sigma_{p_n^*}\sqrt{\frac{\log\frac{1}{\delta}}{n}}$ when $a = \frac{1}{\sigma_{p_n^*}}\sqrt{\frac{\log\frac{1}{\delta}}{n}}$. The lemma then follows from the intermediate value theorem.

Consider the construction of $p_n^*$ in Definition 1, and let $r$ be the trimming radius of $p$. We do a case analysis, either $\frac{1}{\sigma_{p_n^*}}\sqrt{\frac{\log\frac{1}{\delta}}{n}} \le 1/r$ or $\frac{1}{\sigma_{p_n^*}}\sqrt{\frac{\log\frac{1}{\delta}}{n}} \ge 1/r$.

**Case** $\frac{1}{\sigma_{p_n^*}}\sqrt{\frac{\log\frac{1}{\delta}}{n}} \le 1/r$**:** At $a = \frac{1}{\sigma_{p_n^*}}\sqrt{\frac{\log\frac{1}{\delta}}{n}}$, we have

$$a \cdot \int_{-\frac{1}{a}}^{\frac{1}{a}} x^2 \, \mathrm{d}p \ge a \cdot \int_{-r}^{r} x^2 \, \mathrm{d}p \quad \text{since } a \le 1/r$$

$$\ge a \cdot \int_{-r}^{r} (x - \mu_{p_n^*})^2 \, \mathrm{d}p \quad \text{since the mean squared error is minimized at the mean}$$

$$\ge \frac{1}{2} a \cdot \sigma_{p_n^*}^2 \quad \text{since over } [-r, r] \text{ we have } \frac{\mathrm{d}p_n^*}{\mathrm{d}p} = \frac{1}{1 - \frac{0.45}{n}\log\frac{1}{\delta}} \le 2 \text{ for suff. small } \frac{\log\frac{1}{\delta}}{n}$$

$$= \frac{1}{2} \frac{1}{\sigma_{p_n^*}} \sqrt{\frac{\log\frac{1}{\delta}}{n}} \cdot \sigma_{p_n^*}^2 \quad \text{by definition of } a$$

$$= \frac{1}{2} \sigma_{p_n^*} \sqrt{\frac{\log\frac{1}{\delta}}{n}}$$

**Case** $\frac{1}{\sigma_{p_n^*}}\sqrt{\frac{\log\frac{1}{\delta}}{n}} \ge 1/r$**:** At $a = \frac{1}{\sigma_{p_n^*}}\sqrt{\frac{\log\frac{1}{\delta}}{n}}$, we have

$$\int_{-\infty}^{-\frac{1}{a}} (-x) \, \mathrm{d}p + \int_{\frac{1}{a}}^{\infty} x \, \mathrm{d}p \ge \int_{-\infty}^{-r} (-x) \, \mathrm{d}p + \int_{r}^{\infty} x \, \mathrm{d}p \quad \text{since } a \ge \frac{1}{r}$$

$$\ge r \cdot \int_{\mathbb{R}\setminus[-r,r]} 1 \, \mathrm{d}p$$

$$= r \cdot \frac{0.45\log\frac{1}{\delta}}{n} \quad \text{by the definition of } p_n^* \text{ and } r$$

$$= 0.45 r \cdot a \cdot \sigma_{p_n^*} \sqrt{\frac{\log\frac{1}{\delta}}{n}}$$

$$\ge 0.45 \sigma_{p_n^*} \sqrt{\frac{\log\frac{1}{\delta}}{n}} \quad \text{since } a \ge \frac{1}{r}$$

Summarizing, in either case, at $a = \frac{1}{\sigma_{p_n^*}}\sqrt{\frac{\log\frac{1}{\delta}}{n}}$, we have that the first moment shift between $q^+$ and $p$ is

$$\int_{-\infty}^{-\frac{1}{a}} (-x) \, \mathrm{d}p + a \cdot \int_{-\frac{1}{a}}^{\frac{1}{a}} x^2 \, \mathrm{d}p + \int_{\frac{1}{a}}^{\infty} x \, \mathrm{d}p \ge \min\left(\frac{1}{2}, 0.45\right) \sigma_{p_n^*} \sqrt{\frac{\log\frac{1}{\delta}}{n}} \ge \frac{1}{8} \sigma_{p_n^*} \sqrt{\frac{\log\frac{1}{\delta}}{n}}$$

yielding the lemma. $\qquad\square$

### B.1 Bounding the squared Hellinger distance

We restate Lemmas 7 and 8 from Section 2.2 for clarity, then give the proofs of them below.

**Lemma 7** *Suppose there is a sufficiently small constant that upper bounds both $\frac{\log\frac{1}{\delta}}{n}$ and $\delta$. Given a distribution $p$, if we construct $q$ as in Case 1 (the large $|\mu_{p_n^*} - \mu_p|$ case) in Definition 4, then $\log(1 - d_H^2(p, q)) \ge \frac{1}{2n} \log 4\delta$.*

**Proof of Lemma 7.** Since Case 1 in the construction of $q$ in Definition 4 linearly interpolates between $p$ and $p_n^*$, with interpolation constant $\lambda = \frac{3}{4}$, thus the Hellinger distance between $p$ and $q$ can be exactly computed as a function of 1) $\frac{0.45\log\frac{1}{\delta}}{n}$, which determines the masses of $p, q$ inside and outside of $p$'s trimming interval, along with 2) the ratio $\frac{\mathrm{d}q}{\mathrm{d}p}$ inside and outside of $p$'s trimming interval, which also depends only on $\frac{0.45}{n}$ and $\lambda$. We then bound this (essentially) univariate expression.

Without loss of generality, assume that $\mu_p = 0$, and let the support of $p_n^*$ be $[-r, r]$. Recall the construction of $q$ in Case 1 of Definition 4: $q = \lambda p + (1 - \lambda)p_n^*$. At $x \notin [-r, r]$, we have $\frac{dq}{dp}(x) = \lambda$. Otherwise, at $x \in [-r, r]$, we have $\frac{dq}{dp}(x) = \frac{d}{dp}\left(\lambda p + \frac{1-\lambda}{1 - \frac{0.45\log \frac{1}{\delta}}{n}}p\right)(x)$ by the definition of $p_n^*$, which is in turn equal to $(1 - \lambda \cdot \frac{0.45\log \frac{1}{\delta}}{n})/(1 - \frac{0.45\log \frac{1}{\delta}}{n})$.

Given the above equalities, we can explicitly calculate $1 - d_H^2(p, q)$ as follows.

$$
\begin{aligned}
1 - d_H^2(p, q) &= \int \sqrt{dp\,dq} \\
&= \int_{\mathbb{R}\setminus[-r,r]} \sqrt{dp\,dq} + \int_{[-r,r]} \sqrt{dp\,dq} \\
&= \int_{\mathbb{R}\setminus[-r,r]} \sqrt{\lambda}\,dp + \int_{[-r,r]} \sqrt{\frac{1 - \lambda \cdot \frac{0.45\log \frac{1}{\delta}}{n}}{1 - \frac{0.45\log \frac{1}{\delta}}{n}}}\,dp \\
&= \sqrt{\lambda} \cdot \frac{0.45\log \frac{1}{\delta}}{n} + \sqrt{\frac{1 - \lambda \cdot \frac{0.45\log \frac{1}{\delta}}{n}}{1 - \frac{0.45\log \frac{1}{\delta}}{n}} \cdot \left(1 - \frac{0.45\log \frac{1}{\delta}}{n}\right)} \qquad \text{by the definition of } p_n^* \\
&= \sqrt{\lambda} \cdot \frac{0.45\log \frac{1}{\delta}}{n} + \sqrt{\left(1 - \lambda \cdot \frac{0.45\log \frac{1}{\delta}}{n}\right)\left(1 - \frac{0.45\log \frac{1}{\delta}}{n}\right)}
\end{aligned}
$$

We now show a technical lemma to lower bound the quantity in the last line.

**Lemma 24** *For any $\lambda \in [\frac{3}{4}, 1]$ and $\beta \in [0, 1]$, we have*

$$
\sqrt{\lambda}\beta + \sqrt{(1 - \lambda\beta)(1 - \beta)} \geq e^{(\lambda-1)\beta}
$$

**Proof.** Take the second derivative of the left hand side with respect to $\beta$, giving $-\frac{(\lambda-1)^2}{4(1-\beta)^{3/2}(1-\lambda\beta)^{3/2}}$, which is negative. On the other hand, the right hand side $e^{(\lambda-1)\beta}$ is an exponential in $\beta$ and hence convex in $\beta$. Therefore, left hand side minus right hand side is concave, meaning that the difference is minimized at either $\beta = 0$ or $\beta = 1$. At $\beta = 0$, both sides are equal to 1. At $\beta = 1$, the left hand side is $\sqrt{\lambda}$ whereas the right hand side is $e^{\lambda-1}$. The inequality $\sqrt{\lambda} \geq e^{\lambda-1}$ is true for any $\lambda \in [\frac{3}{4}, 1]$. $\square$

Using this lemma, we have shown that

$$
\begin{aligned}
\log(1 - d_H^2(p, q)) &= \log\left(\sqrt{\lambda} \cdot \frac{0.45\log \frac{1}{\delta}}{n} + \sqrt{\left(1 - \lambda \cdot \frac{0.45\log \frac{1}{\delta}}{n}\right)\left(1 - \frac{0.45\log \frac{1}{\delta}}{n}\right)}\right) \\
&\geq (\lambda - 1)\frac{0.45\log \frac{1}{\delta}}{n} = (1 - \lambda)\frac{4.5 \log \delta}{10n} \\
&\geq (1 - \lambda)\frac{0.9 \log 4\delta}{n} \qquad \text{since } \delta \text{ is sufficiently small} \\
&\geq \frac{1}{2n} \log 4\delta \qquad \text{by the definition of } \lambda = 3/4 \text{ and that } \log 4\delta < 0
\end{aligned}
$$

$\square$

**Lemma 8** *Suppose there is a sufficiently small constant that upper bounds both $\frac{\log \frac{1}{\delta}}{n}$ and $\delta$. Given a distribution $p$, if we construct $q$ as in Case 2 (the small $|\mu_{p_n^*} - \mu_p|$ case) in Definition 4, then $\log(1 - d_H^2(p, q)) \geq \frac{1}{2n} \log 4\delta$.*

**Proof of Lemma 8.** Recall that given a distribution $p$, in Case 2 of Definition 4, we construct $q$ by picking one of the two non-unit measures $q^+$ and $q^-$ which has mass $\frac{1}{b} \geq 1$. Without loss of generality (via an appropriate reflection of $p$ with respect to $\mu_p$), let $q^+$ be this non-unit measure,

then use $q = bq^+$. We use $a$ to denote the corresponding solution to the equation in Definition 4 (note that $a > 0$).

To relate $d_H^2(p, q)$ to $d_H^2(p, q^+)$, we will need to use the following lemma concerning the generalization of squared Hellinger distance between a distribution and a non-negative measure with mass bigger than 1.

**Lemma 25** *Given a distribution $p$, and a non-negative measure $q$ with $\frac{1}{b} \geq 1$ probability mass, define the (extended) squared Hellinger distance as $d_H^2(p, q) = \frac{1}{2} \int (\sqrt{dp} - \sqrt{dq})^2$. Then, we have*

$$d_H^2(p, q) \geq d_H^2(p, bq)$$

**Proof.** For any $b' \geq 0$, we have

$$\frac{1}{2} d_H^2(p, b'q) = \frac{1}{2} \int (\sqrt{dp} - \sqrt{b'dq})^2 = \frac{1}{2} \left( \int 1 \, dp - 2\sqrt{b'} \int \sqrt{dp \, dq} + b' \int 1 \, dq \right)$$

$$= \frac{1}{2} \left( 1 - 2\sqrt{b'} \int \sqrt{dp \, dq} + \frac{b'}{b} \right)$$

The derivative in $b'$ is therefore

$$\frac{1}{2} \left( \frac{1}{b} - \frac{\int \sqrt{dp \, dq}}{\sqrt{b'}} \right)$$

The derivative is greater than 0 if and only if

$$\sqrt{\frac{b'}{b}} \geq \int \sqrt{dp \, d(bq)}$$

The right hand side is the Bhattacharya coefficient between two distributions, $p$ and $bq$, and hence is upper bounded by 1 as a standard fact. The left hand side on the other hand is at least 1 for all $b' \in [b, 1]$. Therefore, $d_H^2(p, b'q)$ is an increasing function in $b'$ for the range $b' \in [b, 1]$, meaning that $d_H^2(p, q) \geq d_H^2(p, bq)$, as desired. $\square$

We can now upper bound the squared Hellinger distance as follows.

$$d_H^2(p, q) = d_H^2(p, bq^+)$$

$$\leq d_H^2(p, q^+) \quad \text{by Lemma 25, since } \frac{1}{b} \geq 1$$

$$= \frac{1}{2} \int_{-\infty}^{\infty} (\sqrt{dp} - \sqrt{dq^+})^2$$

$$= \frac{1}{2} \int_{-\infty}^{\infty} \left( 1 - \sqrt{1 + \min(1, \max(-1, ax))} \right)^2 \, dp \quad \text{by definition of } q^+$$

$$\leq \frac{1}{2} \int_{-\infty}^{\infty} \min(1, (ax)^2) \, dp \quad \text{since the inequality holds pointwise}$$

$$= \frac{1}{2} \left( \int_{-\infty}^{-\frac{1}{a}} 1 \, dp + a^2 \int_{-\frac{1}{a}}^{\frac{1}{a}} x^2 \, dp + \int_{\frac{1}{a}}^{\infty} 1 \, dp \right)$$

$$= \frac{a}{2} \cdot \left( \int_{-\infty}^{-\frac{1}{a}} \frac{1}{a} \, dp + a \int_{-\frac{1}{a}}^{\frac{1}{a}} x^2 \, dp + \int_{\frac{1}{a}}^{\infty} \frac{1}{a} \, dp \right)$$

$$\leq \frac{a}{2} \cdot \left( \int_{-\infty}^{-\frac{1}{a}} (-x) \, dp + a \int_{-\frac{1}{a}}^{\frac{1}{a}} x^2 \, dp + \int_{\frac{1}{a}}^{\infty} x \, dp \right)$$

$$= \frac{a}{2} \cdot \frac{1}{8} \cdot \sigma_{p_n^*} \sqrt{\frac{\log \frac{1}{\delta}}{n}} \quad \text{since } a \text{ satisfies the equation of Definition 4}$$

$$\leq \sqrt{\frac{\log \frac{1}{\delta}}{n}} \cdot \frac{1}{\sigma_{p_n^*}} \cdot \frac{1}{16} \cdot \sigma_{p_n^*} \sqrt{\frac{\log \frac{1}{\delta}}{n}} \quad \text{since } a \text{ is upper bounded by Definition 4}$$

$$= \frac{\log \frac{1}{\delta}}{16n} \leq \frac{\log \frac{1}{4\delta}}{4n} \quad \text{since } \delta \text{ is sufficiently small}$$

Observe that for sufficiently small $z > 0$, we have $\log(1 - z) \geq -2z$. Since $\delta$ and $\frac{\log \frac{1}{\delta}}{n}$ are assumed to be sufficiently small, we have

$$\log(1 - d_{\mathrm{H}}^2(p, q)) \geq -\frac{\log \frac{1}{4\delta}}{2n} = \frac{1}{2n} \log 4\delta$$

$\square$

## C   Error analysis for median-of-means

In this section, we present the matching upper bound result of the performance of the median-of-means estimator. We restate the estimator and the proposition for the sake of clarity.

---

**Algorithm 1** Standard Median-of-Means Estimator

---

Inputs: $n$ independent samples $\{x_i\}$ from an unknown distribution $p$; and confidence parameter $\delta$

1. Divide the samples into $4.5\log \frac{1}{\delta}$ groups with equal size.
2. Compute the mean of each group.
3. Return the median of these $4.5\log \frac{1}{\delta}$ means.

---

**Proposition 14** *Consider a distribution $p$ with mean $\mu_p$, a sample size $n$, and a median-of-means group count $4.5\log \frac{1}{\delta}$. Let $p_n^*$ be the $\frac{0.45}{n}\log \frac{1}{\delta}$-trimmed distribution from Definition 1, and $\mu_{p_n^*}$ and $\sigma_{p_n^*}$ be the mean and standard deviation of $p_n^*$ respectively. Then, the median-of-means estimator has error $|\mu_p - \mu_{p_n^*}| + 3\sigma_{p_n^*}\sqrt{\frac{4.5\log \frac{1}{\delta}}{n}}$ except with probability at most $\delta$.*

To prove this proposition, we use the following lemma:

**Lemma 26** *Defining $p_n^*$ to be the $\frac{0.45}{n}\log \frac{1}{\delta}$-trimmed version of $p$, then the empirical mean of $n' = \frac{n}{4.5\log \frac{1}{\delta}}$ samples from $p$ is within $3\frac{\sigma_{p_n^*}}{\sqrt{n'}}$ of $\mu_{p_n^*}$, except with probability at most $\frac{1}{5}$.*

**Proof.** Recall $p_n^*$ is $p$ but with $\frac{0.45}{n}\log \frac{1}{\delta}$ probability mass trimmed (and then scaled up to have total mass 1). Thus the probability that any of the $n' = \frac{n}{4.5\log \frac{1}{\delta}}$ samples from $p$ are not in the support of $p_n^*$ is at most $\frac{1}{10}$ by the union bound.

Conditioned on the event stated above not happening, we can view the sampling process as drawing $n'$ samples from $p_n^*$, with mean $\mu_{p_n^*}$ and standard deviation $\sigma_{p_n^*}$. The standard deviation of the *mean* of these $n'$ samples is thus $\frac{\sigma_{p_n^*}}{\sqrt{n'}}$. By the Chebyshev inequality, the probability of the sample mean being more than 3 times its standard deviation from the true mean is at most $\frac{1}{9}$. Thus the empirical mean is within $3\frac{\sigma_{p_n^*}}{\sqrt{n'}}$ of $\mu_{p_n^*}$, except with probability at most $\frac{1}{9}$.

Combining these two case, where the event happens and where it does not, the overall probability of the empirical mean of $n'$ samples being more than $3\frac{\sigma_{p_n^*}}{\sqrt{n}}$ away from the true mean $\mu_{p_n^*}$ is at most $\frac{1}{10} + \frac{9}{10} \cdot \frac{1}{9} = \frac{1}{5}$. $\square$

**Proof of Proposition 14.** Substituting in $n' = \frac{n}{4.5\log \frac{1}{\delta}}$ to Lemma 26 for each group of the estimator, and with the fact that the mean of $p$ is $\mu_p$, we arrive at the conclusion that median-of-means will have error $|\mu_p - \mu_{p_n^*}| + 3\sigma_{p_n^*}\sqrt{\frac{4.5\log \frac{1}{\delta}}{n}}$ except when at least half of the $4.5\log \frac{1}{\delta}$ groups have error $> 3\sigma_{p_n^*}\sqrt{\frac{4.5\log \frac{1}{\delta}}{n}}$. Since the probability of each mean having a big error is at most $\frac{1}{5}$ by Lemma 26, the overall failure probability is thus at most the probability that $4.5\log \frac{1}{\delta}$ coins each of bias $\frac{1}{5}$ will yield majority heads. Letting $k = 4.5\log \frac{1}{\delta}$, we bound this probability with a Chernoff bound, which

for our choice of $t \geq 0$, yields $(\frac{4}{5}e^0 + \frac{1}{5}e^t)^k e^{-\frac{tk}{2}}$. The upper bound is minimized at $t = \log 4$, attaining its minimum of $(\frac{4}{5})^k$. Since $k = 4.5\log\frac{1}{\delta}$, this probability is less than $\delta$. $\qquad\square$

## D  Definition of instance optimality and admissibility

This paper needs a suitable notion of optimality beyond the worst case. One natural definition to consider is the notion of *instance optimality* (see Definition 27 in Appendix D) from the computer science literature [FLN01], which intuitively states that "our algorithm performs at least as well on any instance $p$ as *any* algorithm customized to $p$". However, it is immediate that *no* algorithm $A$ can satisfy such a definition in our setting—for every distribution $p$, there is a trivial estimator that is hardcoded to output the mean $\mu_p$ without looking at any data; and this hardcoded estimator beats any other estimator $A$. On the other hand, the statistics literature commonly uses a different natural notion called *admissibility* (see Definition 28 in Appendix D, also analogous to the economics notion of *Pareto efficiency*), which states that "no algorithm can perform at least as well as our algorithm, and strictly outperforms our algorithm on some instance." While instance optimality is impossible to satisfy, admissibility has the dual problem of being somewhat too weak and too easy to satisfy: a trivial estimator which outputs a hardcoded mean estimate—ignoring any samples—is admissible.

In this short appendix, we give the formal definitions of instance optimality and admissibility for mean estimation over $\mathbb{R}$.

Recall the notation $\mathcal{P}_1$ for the set of distributions over $\mathbb{R}$ with a finite mean.

**Definition 27 ($\kappa$-Instance Optimality in Mean Estimation)** *For a parameter $\kappa > 1$, sample complexity $n$ and failure probability $\delta$, a mean estimator $\hat{\mu}$ whose error function is $\epsilon_{n,\delta}$ is $\kappa$-instance optimal if, for any distribution $p \in \mathcal{P}_1$, every estimator $\hat{\mu}'$ has error at least $\frac{1}{\kappa}\epsilon_{n,\delta}(p)$ with probability $1 - \delta$ over $n$ i.i.d. samples from $p$.*

As we remarked in the introduction, there is no instance optimal mean estimator for any $\kappa > 0$, since for any estimator $\hat{\mu}$, we can find a distribution $p$ on which it has some nonzero error, and thus $\hat{\mu}$ cannot be instance optimal, because it performs infinitely worse on $p$ in comparison with the trivial "hardcoded" estimator that always outputs $\mu_p$ without looking at any samples.

**Definition 28 (Admissibility in Mean Estimation)** *For sample complexity $n$ and failure probability $\delta$, a mean estimator $\hat{\mu}$ whose error as a function of the distribution $p$ is $\epsilon_{n,\delta}(p)$, is called "admissible" if there is* no *estimator $\hat{\mu}'$ with error function $\epsilon'_{n,\delta}$ such that*

- *For every distribution $p \in \mathcal{P}_1$, $\epsilon'_{n,\delta}(p) \leq \epsilon_{n,\delta}(p)$*

- *There exists a distribution $p^* \in \mathcal{P}_1$ such that $\epsilon'_{n,\delta}(p^*) < \epsilon_{n,\delta}(p^*)$*

## E  Hardcoded estimators are not neighborhood optimal

We perform the basic "sanity check" for the neighborhood structure of Definition 15, and formally show that under this neighborhood definition, no trivial hardcoded estimator is $\kappa$-neighborhood optimal for any $\kappa > 1$.

**Proposition 29** *Consider a mean estimator $\hat{\mu}_\beta$ which ignores any of its inputs and always outputs the value $\beta$ for some $\beta \in \mathbb{R}$. For any parameter $\kappa > 1$, $\hat{\mu}_\beta$ cannot be $\kappa$-neighborhood optimal with respect to $N_{n,\delta}$ defined in Definition 15.*

**Proof.** For the sake of contradiction, suppose $\hat{\mu}_\beta$ is $\kappa$-neighborhood optimal. The error function $\epsilon_\beta$ for $\hat{\mu}_\beta$ is simply $\epsilon_\beta(p) = |\mu_p - \beta|$. Since $\hat{\mu}_\beta$ is $\kappa$-neighborhood optimal, there must exist some error function $\epsilon^{\text{pareto}} \geq \frac{1}{\kappa}\epsilon_\beta$ such that $\epsilon^{\text{pareto}}$ is a neighborhood Pareto bound with respect to $N_{n,\delta}$. We will reach a contradiction by showing that $\epsilon^{\text{pareto}}$ cannot be a neighborhood Pareto bound.

Pick an arbitrary distribution $p$ whose mean $\mu_p$ is $\beta + (1 + \kappa^2) \cdot \epsilon_{n,\delta}(p)$. This is always possible since $\epsilon_{n,\delta}(p)$ is translation-invariant in $p$. By Property 3 of Definition 15 and the reverse triangle inequality, any $q \in N_{n,\delta}(p)$ has mean $\mu_q$ such that $\epsilon_\beta(q) = |\mu_q - \beta| \geq \kappa^2\epsilon_{n,\delta}(p)$. Since $\epsilon^{\text{pareto}}$

is a neighborhood Pareto bound, there must be no estimator $\tilde{\mu}$ such that for every distribution $q \in N_{n,\delta}(p) \cup \{p\}$, $|\tilde{\mu} - \mu_q| < \kappa \epsilon_{n,\delta}(p)$ with probability $1 - \delta$ over $n$ samples from $q$; otherwise, we would have $|\tilde{\mu} - \mu_q| < \kappa \epsilon_{n,\delta}(p) \leq \frac{1}{\kappa}|\mu_q - \beta| = \frac{1}{\kappa}\epsilon_\beta(q) \leq \epsilon^{\text{pareto}}(q)$, which contradicts $\epsilon^{\text{pareto}}$ being a neighborhood Pareto bound. However, we can simply pick $\tilde{\mu} = \hat{\mu}_{\mu_p}$, the trivial estimator that outputs $\mu_p$ always. Again by Property 3 of Definition 15, we have for every $q \in N_{n,\delta}(p) \cup \{p\}$ that $|\mu_q - \tilde{\mu}| = |\mu_q - \mu_p| \leq \epsilon_{n,\delta}(p) < \kappa \epsilon_{n,\delta}(p)$ for $\kappa > 1$. We have thus reached the desired contradiction. $\qquad\square$

# F    Remaining proofs for Lemma 17

In Section 2, we have already shown that the construction of $q$ as in Definition 4 satisfies properties 2 and 4 of Definition 15. To complete the proof of Lemma 17, we show that $q$ satisfies property 1 in Appendix F.1, and property 3 in Appendix F.2.

## F.1    Upper bounding $|\mu_q - \mu_p|$

We show in this section that, for both cases in Definition 4, the construction of $q$ is such that $|\mu_q - \mu_p|$ is appropriately upper bounded. We will show that $|\mu_q - \mu_p| \leq \epsilon_{n,\delta}(p)$, as part of the proof showing that $q \in N_{n,\delta}(p)$. We furthermore show other upper bounds of $|\mu_q - \mu_p|$, for example, that $|\mu_q - \mu_p| \leq \frac{r}{4}$ if $r$ is the trimming radius for the construction of $p_n^*$ from $p$, which will be useful for the later sections.

**Lemma 30** *Assuming $p$ has mean 0, and is trimmed to the interval $[-r, r]$ when constructing $p_n^*$ (as in Definition 1), then Case 1 (the large $|\mu_{p_n^*} - \mu_p|$ case) of Definition 4 outputs a distribution $q$ such that $|\mu_q - \mu_p| \leq \frac{r}{4}$ and $|\mu_q - \mu_p| \leq \frac{1}{4}\epsilon_{n,\delta}(p)$.*

**Proof.** The distribution $q$ is constructed to be a convex combination of $p$ and $p_n^*$, with interpolation parameter $1 - \lambda$ equal to $\frac{1}{4}$ by definition. Thus $|\mu_q - \mu_p| \leq \frac{1}{4}|\mu_{p_n^*} - \mu_p|$, where this last quantity is at most $\frac{r}{4}$ since $p_n^*$ is supported on $[-r, r]$ and thus can have mean at most distance $r$ away from the origin. Furthermore, by the definition of $\epsilon_{n,\delta}(p)$, we have $|\mu_q - \mu_p| \leq \frac{1}{4}|\mu_{p_n^*} - \mu_p| \leq \frac{1}{4}\epsilon_{n,\delta}(p)$. $\quad\square$

**Lemma 31** *Assuming $p$ has mean 0, and is trimmed to the interval $[-r, r]$ when constructing $p_n^*$ (as in Definition 1), then Case 2 (the small $|\mu_{p_n^*} - \mu_p|$ case) of Definition 4 outputs a distribution $q$ such that $|\mu_q - \mu_p| \leq \frac{1}{8}\sigma_{p_n^*}\sqrt{\frac{\log\frac{1}{\delta}}{n}}$. As a corollary, this is further upper bounded by $\epsilon_{n,\delta}(p)$.*

*Additionally, if $\frac{\log\frac{1}{\delta}}{n}$ is upper bounded by some sufficiently small absolute constant, then $\frac{1}{8}\sigma_{p_n^*}\sqrt{\frac{\log\frac{1}{\delta}}{n}}$ can also be bounded by $\frac{r}{4}$.*

**Proof.** Without loss of generality, suppose $q^+$ is the non-unit measure constructed by Case 2 of Definition 4 which will be downscaled to create $q$. We know from Lemma 23 that $\left|\int_{-\infty}^{\infty} x \, dq^+\right| = \left|\int_{-\infty}^{\infty} x(dq^+ - dp)\right| = \frac{1}{8}\sigma_{p_n^*}\sqrt{\frac{\log\frac{1}{\delta}}{n}}$. Since $q$ is a downscale of $q^+$, we then arrive at $|\mu_q| \leq \left|\int_{-\infty}^{\infty} x \, dq^+\right| = \frac{1}{8}\sigma_{p_n^*}\sqrt{\frac{\log\frac{1}{\delta}}{n}}$. By the definition of $\epsilon_{n,\delta}(p)$, we have $|\mu_q - \mu_p| \leq \frac{1}{8}\sigma_{p_n^*}\sqrt{\frac{\log\frac{1}{\delta}}{n}} \leq \epsilon_{n,\delta}(p)$.

For the other corollary, observe that since $p_n^*$'s support is in $[-r, r]$, $\sigma_{p_n^*}$ is then upper bounded by $2r$. Therefore, under the assumption that $\frac{\log\frac{1}{\delta}}{n}$ is bounded by a sufficiently small constant, we have $\frac{1}{8}\sigma_{p_n^*}\sqrt{\frac{\log\frac{1}{\delta}}{n}} \leq \frac{r}{4}$. $\quad\square$

## F.2    Showing that $\epsilon_{n/3,\delta}(q) \leq O(\epsilon_{n,\delta}(p))$

Since $\epsilon_{n/3,\delta}(q) = O\left(|\mu_q - \mu_{q_{n/3}^*}| + \sigma_{q_{n/3}^*}\sqrt{\frac{\log\frac{1}{\delta}}{n}}\right)$, we will separately show that $|\mu_q - \mu_{q_{n/3}^*}| = O(\epsilon_{n,\delta}(p))$ (Lemma 35) and $\sigma_{q_{n/3}^*}\sqrt{\frac{\log\frac{1}{\delta}}{n}} = O(\epsilon_{n,\delta}(p))$ (Lemma 36).

For the second item, the proof of Lemma 36 is relatively self-contained, though requiring some minute calculations. On the other hand, the analysis of $|\mu_q - \mu_{q^*_{n/3}}|$ is non-trivial. To upper bound this term, we consider an intermediate (non-unit) measure $\tilde{q}$ resulting from trimming $q$ to the *same* interval as $p^*_n$. We then show Lemma 32 which bounds the difference between first moments of $q^*_{n/3}$ and $\tilde{q}$, and Lemmas 33 and 34 which bound the difference between the first moments of $q$ and $\tilde{q}$ in the two cases of the construction of $q$. The combination of these three lemmas gives Lemma 35, which bounds $|\mu_q - \mu_{q^*_{n/3}}|$ as desired.

In much of the analysis in this section, we will assume without loss of generality that $p$ has mean 0, is trimmed to the unit interval $[-1, 1]$ when constructing $p^*_n$.

**Lemma 32** *Let $n$ be the number of samples and $\delta$ be the failure probability, and suppose there is a sufficiently small absolute constant which upper bounds $\frac{\log \frac{1}{\delta}}{n}$. Given a distribution $p$ with $\mu_p = 0$ and whose trimming radius for constructing $p^*_n$ is equal to $1$, suppose the distribution $q$ is such that $\frac{dq}{dp} \le 2$ and $|\mu_q| \le \frac{1}{4}$. Letting $q^*_{n/3}$ be the $(\frac{1.35}{n}\log \frac{1}{\delta})$-trimmed version of distribution $q$, trimming to some interval $[\mu_q - r_q, \mu_q + r_q]$, then:*

- *When $[\mu_q - r_q, \mu_q + r_q]$ is a subset of $[-1, 1]$ then*

$$\left| \int_{-1}^{1} (x - \mu_q)\, dq - \int_{\mu_q - r_q}^{\mu_q + r_q} (x - \mu_q)\, dq \right| \le |\mu_p - \mu_q| + \sqrt{\frac{3}{5}}\sigma_{p^*_n}\sqrt{\frac{4.5\log \frac{1}{\delta}}{n}}$$

- *When $[\mu_q - r_q, \mu_q + r_q]$ extends outside $[-1, 1]$ then*

$$\left| \int_{-1}^{1} (x - \mu_q)\, dq - \int_{\mu_q - r_q}^{\mu_q + r_q} (x - \mu_q)\, dq \right| \le \frac{2.8125\log \frac{1}{\delta}}{n} \le 2\epsilon_{n,\delta}(p)$$

**Proof of the first bullet.** Define $S = [-1, 1] \setminus [\mu_q - r_q, \mu_q + r_q]$. Firstly, due to triangle inequality, we have

$$\left| \int_S (x - \mu_q)\, dq \right| \le \left| \int_S (\mu_p - \mu_q)\, dq \right| + \left| \int_S (x - \mu_p)\, dq \right| \le |\mu_p - \mu_q| + \left| \int_S (x - \mu_p)\, dq \right|$$

The Cauchy-Schwarz inequality then says that

$$\left| \int_S (x - \mu_p)\, dq \right| \le \sqrt{\int_S (x - \mu_p)^2\, dq}\sqrt{\int_S dq}$$

Note that the second integral in the right hand side is bounded by the amount of trimmed probability mass, which is at most $\frac{1.35\log \frac{1}{\delta}}{n}$. The first integral, since $\frac{dq}{dp} \le 2$, is bounded by twice the variance of $p$ in $[-1, 1]$, namely $2\sigma_{p^*_n}{}^2$. Thus the product of the square roots of the two integrals on the right hand side is bounded by $\sigma_{p^*_n}\sqrt{2\frac{1.35\log \frac{1}{\delta}}{n}}$, yielding the desired bound. $\qquad\square$

**Proof of the second bullet.** Without loss of generality, we assume $\mu_q \ge \mu_p = 0$. Thus by the assumption of this case, that $[\mu_q - r_q, \mu_q + r_q]$ extends outside $[-1, 1]$, we have that $\mu_q + r_q > 1$, because of the asymmetry that $\mu_q \ge 0$. Since $\mu_q \le \frac{1}{4}$ by the lemma assumption, we also have $r_q \ge \frac{3}{4}$.

Recall that $p$ has a total of $\frac{0.45\log \frac{1}{\delta}}{n}$ mass outside $[-1, 1]$ by the lemma assumption, and that $\frac{dq}{dp} \le 2$. This implies that $q$ has at most $\frac{0.9\log \frac{1}{\delta}}{n}$ mass outside $[-1, 1]$. As $\frac{1.35\log \frac{1}{\delta}}{n}$ mass is trimmed from $q$ to construct $q^*_{n/3}$, there is thus at least $\frac{0.45\log \frac{1}{\delta}}{n}$ mass trimmed from $q$ *inside* the interval $[-1, 1]$. Furthermore, since $\frac{dq}{dp} \le 2$, we conclude that there is thus at least $\frac{0.225\log \frac{1}{\delta}}{n}$ mass trimmed from $p$ inside the interval $[-1, 1]$. Now denote the set on which $q$ is trimmed, restricted to $[-1, 1]$ by $S = [-1, 1] \setminus [\mu_q - r_q, \mu_q + r_q]$. Since $p$ has at least $\frac{0.225\log \frac{1}{\delta}}{n}$ mass in $S$, meaning that $S \neq \emptyset$, and $\mu_q + r_q > 1$ from earlier, it must thus be the case that $\mu_q - r_q > -1$ and therefore $r_q < \frac{5}{4}$.

Since $\min_{x_i \in S} |x_i| > \frac{1}{2}$, we conclude that $(\sigma_{p_n^*})^2 + (\mu_{p_n^*})^2 = \int_{-1}^{1} x^2 \, dp_n^* \geq \int_{-1}^{1} x^2 \, dp \geq \int_S x^2 \, dp \geq \frac{c}{2^2 \cdot 20} \frac{\log \frac{1}{\delta}}{n}$. Using the standard inequality that the $\ell_1$ norm is at least the $\ell_2$ norm, we have that $\sigma_{p_n^*} + \mu_{p_n^*} \geq \sqrt{(\sigma_{p_n^*})^2 + (\mu_{p_n^*})^2} \geq \sqrt{\frac{c}{2^2 \cdot 20} \frac{\log \frac{1}{\delta}}{n}}$. Thus, since $\sqrt{\frac{4.5 \log \frac{1}{\delta}}{n}} \leq \frac{1}{3}$ by the lemma assumption, we have that $\epsilon_{n,\delta}(p) = \sigma_{p_n^*} \sqrt{\frac{4.5 \log \frac{1}{\delta}}{n}} + \mu_{p_n^*} \geq \sqrt{\frac{9}{80}} \frac{4.5 \log \frac{1}{\delta}}{n}$. This yields the second inequality in the second bullet point.

We now prove the first inequality in the second bullet, which bounds the difference between two integrals; we bound this by bounding the probability mass in each integral and multiplying this by a bound on the integrand of each integral. Explicitly, the amount of probability mass of $q$ that is in $[-1, 1]$ but outside of $[\mu_q - r_q, \mu_q + r_q]$ is at most $\frac{1.35 \log \frac{1}{\delta}}{n}$, and furthermore, for every $x \in [-1, 1] \setminus [\mu_q - r_q, \mu_q + r_q]$, we have $|x - \mu_q| \leq \frac{5}{4}$ due to $|\mu_q| \leq \frac{1}{4}$. In parallel, the amount of probability mass of $p$ that is outside of $[-1, 1]$ but in $[\mu_q - r_q, \mu_q + r_q]$ is at most $\frac{0.45 \log \frac{1}{\delta}}{n}$, and since $\frac{dq}{dp} \leq 2$, we conclude that $q$ has at most $\frac{0.9 \log \frac{1}{\delta}}{n}$ mass outside of $[-1, 1]$ but in $[\mu_q - r_q, \mu_q + r_q]$. Also, for every $x \in [\mu_q - r_q, \mu_q + r_q] \setminus [-1, 1]$, we have $|x - \mu_q| \leq \frac{5}{4}$ because $r_q \leq \frac{5}{4}$. Thus, the left hand side of the second bullet is at most $(2 + 3) \frac{5}{4} 0.45 \frac{\log \frac{1}{\delta}}{n} = 2.8125 \frac{\log \frac{1}{\delta}}{n}$, as claimed. $\qquad\square$

**Lemma 33** *Given a distribution $p$ with $\mu_p = 0$ and whose trimming radius for constructing $p_n^*$ is equal to 1, let $q$ be the distribution constructed from $p$ as in Case 2 (the small $|\mu_{p_n^*} - \mu_p|$ case) of Definition 4. Then $\left| \int_{\mathbb{R} \setminus [-1,1]} (x - \mu_q) \, dq \right| \leq 5 \sigma_{p_n^*} \sqrt{\frac{\log \frac{1}{\delta}}{n}}$.*

**Proof.** Notice that

$$\left| \int_{\mathbb{R} \setminus [-1,1]} (x - \mu_q) \, dq \right| \leq \left| \int_{\mathbb{R} \setminus [-1,1]} x \, dq \right| + \left| \int_{\mathbb{R} \setminus [-1,1]} -\mu_q \, dq \right| \leq \left| \int_{\mathbb{R} \setminus [-1,1]} x \, dq \right| + |\mu_q|$$

and since $|\mu_q| \leq \frac{1}{8} \sigma_{p_n^*} \sqrt{\frac{\log \frac{1}{\delta}}{n}}$ by Lemma 31, we only have to bound the first term by some constant multiple of $\sigma_{p_n^*} \sqrt{\frac{\log \frac{1}{\delta}}{n}}$.

Recall the construction of $q$ in Definition 4, as a downscaled version of either $q^+$ or $q^-$. Without loss of generality (by reflecting $p$ and $q$ as appropriate), we assume we use $q^+$. Specifically, $q^+$ is the non-unit measure such that $\frac{dq^+}{dp}(x) = 1 + \min(1, \max(-1, ax))$ for some value $a$ specified in Definition 4. Since $q$ is a downscaled version of $q^+$, we will bound $\left| \int_{\mathbb{R} \setminus [-1,1]} x \, dq \right| \leq \left| \int_{\mathbb{R} \setminus [-1,1]} x \, dq^+ \right|$ by a constant multiple of $\sigma_{p_n^*} \sqrt{\frac{\log \frac{1}{\delta}}{n}}$.

Observe that for any $a > 0$, we have that $x$ and $\min(1, \max(-1, ax))$ have the same sign everywhere, which means

$$\left| \int_{\mathbb{R} \setminus [-1,1]} x \min(1, \max(-1, ax)) \, dp \right| \leq \left| \int_{\mathbb{R}} x \min(1, \max(-1, ax)) \, dp \right|$$

Additionally, because $\frac{\log \frac{1}{\delta}}{n}$ is upper bounded by a sufficiently small absolute constant and $|\mu_{p_n^*}| \leq \sigma_{p_n^*} \sqrt{\frac{4.5}{n} \log \frac{1}{\delta}}$ from the lemma assumption, we get $\left| \frac{\mu_{p_n^*}}{1 - \frac{0.45}{n} \log \frac{1}{\delta}} \right| \leq 2 \sigma_{p_n^*} \sqrt{\frac{0.45}{n} \log \frac{1}{\delta}} \leq 4.5 \sigma_{p_n^*} \sqrt{\frac{\log \frac{1}{\delta}}{n}}$.

We can now upper bound $\left| \int_{\mathbb{R} \setminus [-1,1]} x \, dq^+ \right|$ as follows.

$$\left|\int_{\mathbb{R}\setminus[-1,1]} x\,\mathrm{d}q^+\right| = \left|\int_{\mathbb{R}\setminus[-1,1]} x(1+\min(1,\max(-1,ax)))\,\mathrm{d}p\right|$$

$$\leq \left|\int_{\mathbb{R}\setminus[-1,1]} x\,\mathrm{d}p\right| + \left|\int_{\mathbb{R}\setminus[-1,1]} x\min(1,\max(-1,ax))\,\mathrm{d}p\right|$$

$$\leq \left|\int_{\mathbb{R}\setminus[-1,1]} x\,\mathrm{d}p\right| + \left|\int_{\mathbb{R}} x\min(1,\max(-1,ax))\,\mathrm{d}p\right|$$

$$= \left|\int_{\mathbb{R}\setminus[-1,1]} x\,\mathrm{d}p\right| + \left|\int_{\mathbb{R}} x(\mathrm{d}q^+ - \mathrm{d}p)\right|$$

$$= \left|\frac{\mu_{p_n^*}}{1-\frac{0.45}{n}\log\frac{1}{\delta}}\right| + \left|\int_{\mathbb{R}} x\,\mathrm{d}q^+\right| \quad \text{since } \mu_p = 0$$

$$\leq \left|\frac{\mu_{p_n^*}}{1-\frac{0.45}{n}\log\frac{1}{\delta}}\right| + 2|\mu_q| \quad \text{since } q \text{ is a downscale of } q^+ \text{ by factor at most 2}$$

$$\leq 5\sigma_{p_n^*}\sqrt{\frac{\log\frac{1}{\delta}}{n}} \quad \text{using the prior bounds on the two terms and that } 4.5 + \frac{2}{8} \leq 5$$

$\square$

**Lemma 34** *Given a distribution $p$ with $\mu_p = 0$ and whose trimming radius for constructing $p_n^*$ is equal to 1, let $q$ be the distribution constructed from $p$ as in Case 1 (the large $|\mu_{p_n^*} - \mu_p|$ case) of Definition 4. Then $\left|\int_{\mathbb{R}\setminus[-1,1]}(x-\mu_q)\,\mathrm{d}q\right| \leq 5|\mu_{p_n^*} - \mu_p|$*

**Proof.** From the construction, $q$ outside $[-1,1]$ just consists of the corresponding portion of $p$ multiplied by $\lambda \in (0,1)$. Thus

$$\left|\int_{\mathbb{R}\setminus[-1,1]}(x-\mu_q)\,\mathrm{d}q\right| = \lambda\left|\int_{\mathbb{R}\setminus[-1,1]}(x-\mu_q)\,\mathrm{d}p\right|$$

$$= \lambda\left|(\mu_q - \mu_p)\cdot\frac{0.45\log\frac{1}{\delta}}{n} + \int_{\mathbb{R}\setminus[-1,1]}(x-\mu_p)\,\mathrm{d}p\right|$$

$$\leq |\mu_p - \mu_q| + \frac{|\mu_{p_n^*} - \mu_p|}{1-\frac{0.45\log\frac{1}{\delta}}{n}}$$

Since $q$ is an interpolation between $p$ and $p_n^*$, the mean of $q$ is in between the mean of $p$ and the mean of $p_n^*$, and thus $|\mu_p - \mu_q| \leq |\mu_{p_n^*} - \mu_p|$. Furthermore, since $\frac{1}{n}\log\frac{1}{\delta}$ is upper bounded by some sufficiently small absolute constant, the entire bound above can be finally bounded by $5|\mu_{p_n^*} - \mu_p|$, as desired. $\square$

**Lemma 35** *Given a distribution $p$ with $\mu_p = 0$ and whose trimming radius for constructing $p_n^*$ is equal to 1, consider constructing $q$ according to Definition 4. Recall also the notation for the distribution $q_{n/3}^*$ which is the $(\frac{1.35}{n}\log\frac{1}{\delta})$-trimmed version of $q$ as in Definition 1, and suppose $q_{n/3}^*$ is formed by trimming $q$ to some interval $[\mu_q - r_q, \mu_q + r_q]$. Assuming that $\frac{\log\frac{1}{\delta}}{n}$ is upper bounded by some sufficiently small absolute constant, then $|\mu_q - \mu_{q_{n/3}^*}| \leq 50\epsilon_{n,\delta}(p)$.*

**Proof.** By the definition of $q_{n/3}^*$,

$$\left(1-\frac{1.35\log\frac{1}{\delta}}{n}\right)\mu_q + \frac{1.35\log\frac{1}{\delta}}{n}\mu_q = \mu_q = \left(1-\frac{1.35\log\frac{1}{\delta}}{n}\right)\mu_{q_{n/3}^*} + \int_{\mathbb{R}\setminus[\mu_q-r_q,\mu_q+r_q]} x\,\mathrm{d}q$$

Further noting that $\int_{\mathbb{R}\setminus[\mu_q-r_q,\mu_q+r_q]} \mu_q \, \mathrm{d}q = \frac{1.35\log\frac{1}{\delta}}{n}\mu_q$, since we trim exactly $\frac{1.35\log\frac{1}{\delta}}{n}$ probability mass outside of $[\mu_q - r_q, \mu_q + r_q]$, we can rearrange the above to get

$$\left(1 - \frac{1.35\log\frac{1}{\delta}}{n}\right)(\mu_q - \mu_{q^*_{n/3}}) = \int_{\mathbb{R}\setminus[\mu_q-r_q,\mu_q+r_q]} (x - \mu_q) \, \mathrm{d}q$$

As a result, to bound $|\mu_q - \mu_{q^*_{n/3}}|$, it suffices to bound $\left|\int_{\mathbb{R}\setminus[\mu_q-r_q,\mu_q+r_q]} (x - \mu_q) \, \mathrm{d}q\right|$.

By the triangle inequality, we have that

$$\left|\int_{\mathbb{R}\setminus[\mu_q-r_q,\mu_q+r_q]} (x - \mu_q) \, \mathrm{d}q\right| = \left|\int_{\mathbb{R}\setminus[-1,1]} (x - \mu_q) \, \mathrm{d}q + \int_{-1}^{1} (x - \mu_q) \, \mathrm{d}q - \int_{\mu_q-r_q}^{\mu_q+r_q} (x - \mu_q) \, \mathrm{d}q\right|$$

$$\leq \left|\int_{\mathbb{R}\setminus[-1,1]} (x - \mu_q) \, \mathrm{d}q\right| + \left|\int_{-1}^{1} (x - \mu_q) \, \mathrm{d}q - \int_{\mu_q-r_q}^{\mu_q+r_q} (x - \mu_q) \, \mathrm{d}q\right|$$

and we bound the two terms separately.

By Lemmas 33 and 34, we have $\left|\int_{\mathbb{R}\setminus[-1,1]} (x - \mu_q) \, \mathrm{d}q\right| \leq 5|\mu_p - \mu_{p^*_n}| + 5\sigma_{p^*_n}\sqrt{\frac{\log\frac{1}{\delta}}{n}}$ in either case of the construction of $q$.

To apply Lemma 32, we need to verify that the construction of $q$ is such that $|\mu_q - \mu_p| \leq 1/4$ when the trimming radius for constructing $p^*_n$ from $p$ is 1. This was shown in Lemmas 30 and 31. Therefore, by the two cases of Lemma 32, we have $\left|\int_{-1}^{1} (x - \mu_q) \, \mathrm{d}q - \int_{\mu_q-r_q}^{\mu_q+r_q} (x - \mu_q) \, \mathrm{d}q\right| \leq \max\left(|\mu_p - \mu_q| + \sqrt{\frac{3}{5}}\sigma_{p^*_n}\sqrt{\frac{4.5\log\frac{1}{\delta}}{n}}, 2\epsilon_{n,\delta}(p)\right)$. From Lemmas 30 and 31 again, we have that $|\mu_p - \mu_q| \leq \epsilon_{n,\delta}(p)$.

Combining all these bounds, we have

$$|\mu_q - \mu_{q^*_{n/3}}| \leq \frac{1}{1 - \frac{1.35\log\frac{1}{\delta}}{n}}\left(\left|\int_{\mathbb{R}\setminus[-1,1]} (x - \mu_q) \, \mathrm{d}q\right| + \left|\int_{-1}^{1} (x - \mu_q) \, \mathrm{d}q - \int_{\mu_q-r_q}^{\mu_q+r_q} (x - \mu_q) \, \mathrm{d}q\right|\right)$$

$$\leq 2\left(\left|\int_{\mathbb{R}\setminus[-1,1]} (x - \mu_q) \, \mathrm{d}q\right| + \left|\int_{-1}^{1} (x - \mu_q) \, \mathrm{d}q - \int_{\mu_q-r_q}^{\mu_q+r_q} (x - \mu_q) \, \mathrm{d}q\right|\right)$$

$$\leq 10|\mu_p - \mu_{p^*_n}| + 10\sigma_{p^*_n}\sqrt{\frac{\log\frac{1}{\delta}}{n}} + 2\max\left(\epsilon_{n,\delta}(p) + \sqrt{\frac{3}{5}}\sigma_{p^*_n}\sqrt{\frac{4.5\log\frac{1}{\delta}}{n}}, 2\epsilon_{n,\delta}(p)\right)$$

$$\leq 50\epsilon_{n,\delta}(p)$$

where the last inequality uses the definition of $\epsilon_{n,\delta}(p) = |\mu_p - \mu_{p^*_n}| + \sigma_{p^*_n}\sqrt{\frac{4.5\log\frac{1}{\delta}}{n}}$. $\qquad\square$

**Lemma 36** *Given a distribution $p$ with $\mu_p = 0$ and whose trimming radius for constructing $p^*_n$ is equal to 1, consider any distribution $q$ such that $\frac{\mathrm{d}q}{\mathrm{d}p} \leq 2$ and $|\mu_q| \leq \frac{1}{4}$. Recall the notation for the distribution $q^*_{n/3}$ which is the $(\frac{1.35}{n}\log\frac{1}{\delta})$-trimmed version of $q$ as in Definition 1, and suppose $q^*_{n/3}$ is formed by trimming $q$ to some interval $[\mu_q - r_q, \mu_q + r_q]$. Assuming that $\frac{\log\frac{1}{\delta}}{n}$ is upper bounded by some sufficiently small absolute constant, then $\sigma_{q^*_{n/3}} \leq 50(\sigma_{p^*_n} + |\mu_p - \mu_{p^*_n}|)$.*

**Proof.** Without loss of generality, assume that $\mu_q \geq 0$. In the following proof, we will aim to upper bound $\mathbb{E}_{X\leftarrow q^*}[X^2]$ by $57((\sigma_{p^*_n})^2 + (\mu_{p^*_n})^2)$, which implies $\sigma_{q^*_{n/3}} \leq \sqrt{\mathbb{E}_{X\leftarrow q^*}[X^2]} \leq \sqrt{57}\sqrt{(\sigma_{p^*_n})^2 + (\mu_{p^*_n})^2} \leq 8(\sigma_{p^*_n} + |\mu_{p^*_n}|)$, yielding the lemma statement.

Before we bound $\mathbb{E}_{X\leftarrow q^*}[X^2]$, we first show that by the assumption of $\mu_q \geq 0$, it must be the case that $\mu_q - r_q \geq -1$.

Recall that $p$ has a total of $\frac{0.45\log\frac{1}{\delta}}{n}$ mass outside $[-1,1]$ by the lemma assumption, and that $\frac{dq}{dp} \leq 2$. This implies that $q$ has at most $\frac{0.9\log\frac{1}{\delta}}{n}$ outside $[-1,1]$. Since $\frac{1.35\log\frac{1}{\delta}}{n}$ mass is trimmed from $q$ to construct $q^*_{n/3}$, there is thus at least $\frac{0.45\log\frac{1}{\delta}}{n}$ mass trimmed from $q$ *inside* the interval $[-1,1]$.

Since $\mu_q \geq \mu_p = 0$, it must be the case that $\mu_q - r_q \geq -1$; otherwise $r_q \geq 1$ and hence $\mu_q + r_q \geq 1$, which would mean that $[\mu_q - r_q, \mu_q + r_q]$ completely contains $[-1,1]$, contradicting the fact that $q$ trims non-zero mass from the interval $[-1,1]$.

We will now start bounding $\mathbb{E}_{X \leftarrow q^*}[X^2]$.

$$
\begin{aligned}
\mathop{\mathbb{E}}_{X \leftarrow q^*_{n/3}}[X^2] &= \int_{\mu_q - r_q}^{\mu_q + r_q} x^2\, dq^*_{n/3} \\
&= \frac{1}{1 - \frac{1.35\log\frac{1}{\delta}}{n}} \int_{\mu_q - r_q}^{\mu_q + r_q} x^2\, dq \\
&\leq \frac{2}{1 - \frac{1.35\log\frac{1}{\delta}}{n}} \int_{\mu_q - r_q}^{\mu_q + r_q} x^2\, dp \quad \text{since } \frac{dq}{dp} \leq 2 \\
&\leq \frac{2}{1 - \frac{1.35\log\frac{1}{\delta}}{n}} \int_{-1}^{1} x^2\, dp + \frac{2}{1 - \frac{1.35\log\frac{1}{\delta}}{n}} \int_{1}^{\max(1,\mu_q + r_q)} x^2\, dp
\end{aligned}
$$

We can bound the two integrals separately. First:

$$
\begin{aligned}
\int_{-1}^{1} x^2\, dp &\leq \int_{-1}^{1} x^2\, dp^*_n \quad \text{since } p^*_n \text{ is scaled up after trimming } p \text{ to renormalize} \\
&= (\sigma_{p^*_n})^2 + (\mu_{p^*_n})^2
\end{aligned}
$$

Second, we will bound $\int_{1}^{\max(1,\mu_q+r_q)} x^2\, dp$. If $\mu_q + r_q \leq 1$ then the integral is 0. Otherwise, we have $\mu_q + r_q > 1$, and we have to bound the above integral, using the following bounds on $r_q$, $\mu_q$ and the second moment of $p^*_n$.

Consider the mass of $q$ that was trimmed from within $[-1,1]$. Since $\mu_q + r_q > 1$, the (at least $\frac{0.45\log\frac{1}{\delta}}{n}$) mass that was trimmed from $q$ must be within $[-1, \mu_q - r_q]$. Further recall that $\frac{dq}{dp} \leq 2$, which implies that $p$ has at least $\frac{0.225\log\frac{1}{\delta}}{n}$ mass within $[-1, \mu_q - r_q]$. Since the trimming interval for constructing $p^*_n$ from $p$ is $[-1,1]$, and $p^*_n$ is constructed from scaling up the trimmed version of $p$, we conclude that $p^*_n$ also has at least $\frac{0.225\log\frac{1}{\delta}}{n}$ mass within $[-1, \mu_q - r_q]$.

As we assumed that $\mu_q \leq \frac{1}{4}$ in the lemma statement, we have from $\mu_q + r_q > 1$ that $r_q \geq \frac{3}{4}$. This furthermore implies that $|\mu_q - r_q| \geq \frac{1}{2}$. Thus the $\geq \frac{0.225\log\frac{1}{\delta}}{n}$ probability mass of $p^*_n$ in $[-1, \mu_q - r_q]$ contributes at least $\frac{1}{2^2}\frac{0.225\log\frac{1}{\delta}}{n}$ to the second moment of $p^*_n$; namely $\mathbb{E}_{X \leftarrow p^*_n}[X^2] \geq \frac{0.225}{4}\frac{\log\frac{1}{\delta}}{n}$.

We can also bound $r_q \leq \frac{5}{4}$, since $\mu_q - r_q > -1$ and $\mu_q \leq \frac{1}{4}$; thus $\mu_q + r_q \leq \frac{3}{2}$.

Therefore,

$$\int_1^{\mu_q+r_q} x^2 \, \mathrm{d}p \le (\mu_q + r_q)^2 \int_1^{\mu_q+r_q} \mathrm{d}p$$

$$\le \frac{9}{4} \int_1^{\mu_q+r_q} \mathrm{d}p$$

$$\le \frac{9}{4} \cdot \frac{0.45 \log \frac{1}{\delta}}{n} \quad \text{since } \int_{-1}^1 \mathrm{d}p = 1 - \frac{0.45 \log \frac{1}{\delta}}{n}$$

$$\le 18 \mathop{\mathbb{E}}_{X \leftarrow p_n^*} [X^2] \quad \text{using the prior bound on the second moment}$$

$$= 18((\sigma_{p_n^*})^2 + (\mu_{p_n^*})^2)$$

Summarizing, when $\frac{\log \frac{1}{\delta}}{n}$ is bounded by some sufficiently small absolute constant, we have

$$\mathop{\mathbb{E}}_{X \leftarrow q_{n/3}^*} [X^2] \le \frac{2}{1 - \frac{1.35 \log \frac{1}{\delta}}{n}} \int_{-1}^1 x^2 \, \mathrm{d}p + \frac{2}{1 - \frac{1.35 \log \frac{1}{\delta}}{n}} \int_1^{\max(1, \mu_q + r_q)} x^2 \, \mathrm{d}p$$

$$\le 3 \cdot (1 + 18) \cdot ((\sigma_{p_n^*})^2 + (\mu_{p_n^*})^2)$$

$$= 57((\sigma_{p_n^*})^2 + (\mu_{p_n^*})^2)$$

which yields the lemma, by the argument at the very beginning of the proof.

$\square$

**Lemma 37** *Consider constructing distribution $q$ from $p$ according to Definition 4. Assuming that $\frac{\log \frac{1}{\delta}}{n}$ is upper bounded by some sufficiently small absolute constant, then $\epsilon_{n/3, \delta}(q) \le 100 \epsilon_{n, \delta}(p)$.*

**Proof.** Since the construction of $q$ in Definition 4 satisfies the assumptions of Lemma 36, this lemma follows directly from summing up the bounds of Lemmas 35 and 36, that

$$\epsilon_{n/3, \delta}(q) = |\mu_q - \mu_{q_{n/3}^*}| + \sigma_{q_{n/3}^*} \sqrt{\frac{4.5 \log \frac{1}{\delta}}{n}} \le 50 \epsilon_{n, \delta}(p) + 50(\sigma_{p_n^*} + |\mu_p - \mu_{p_n^*}|) \sqrt{\frac{4.5 \log \frac{1}{\delta}}{n}} \le 100 \epsilon_{n, \delta}(p)$$

where the last inequality uses the definition of $\epsilon_{n, \delta}(p) = |\mu_p - \mu_{p_n^*}| + \sigma_{p_n^*} \sqrt{\frac{4.5 \log \frac{1}{\delta}}{n}}$ and that $\frac{\log \frac{1}{\delta}}{n}$ is bounded by a sufficiently small constant. $\square$

