# OpenReview forum: "Optimality in Mean Estimation: Beyond Worst-Case, Beyond Sub-Gaussian, and Beyond $1+\alpha$ Moments"
_NeurIPS.cc/2023/Conference — NeurIPS 2023 poster_

### Official Review · Reviewer_bzea · 2023-06-12

**Soundness:** 3 good
**Presentation:** 4 excellent
**Contribution:** 3 good
**Rating:** 7
**Confidence:** 4

**Summary:**

The paper studies one-dimensional mean estimation in a beyond-worst case setting.

The main contributions are twofold. First, it gives indistinguishability results which, given a distribution p of finite mean, construct a “partner” distribution q that preserves the moments yet has mean far from p. This implies the subguassian rate via median-of-means is optimal in general. Hence, this result can be viewed as an impossibility theorem against beyond-worst-case analysis for mean estimation.

Second, they give a new framework of neighborhood optimality, and show that median-of-means is optimal up to a constant.


**Strengths:**

The question studied here is well motivated. I believe this is the natural next step following the recent work of Lee-Valiant (2020) and early papers on median-of-means. The paper essentially settles the beyond-worst-case analysis problem in this space (in the negative direction)

This is a quite solid paper at a technical level. The main contribution is interesting and to me somewhat surprising.  I didn’t not check the proof details from the appendix, but reading the main body the main claims appear correct.

Overall the paper is well-written.


**Weaknesses:**

I would be most interested in an extension of the result to the multivariate setting.

Other minor comments below.

By “worst case”, this paper refers to the worst over all distributions (under certain moment constraints). I think this should be remarked early in the introduction, since beyond-worst-case analysis in the TCS community typically refers to non-worst-case data. In our case, the data are drawn from a distribution already. So we are not using this notion in the classic sense.

Abstract line 17:  The construction of q preserves the density of p up to a factor of 2 -> I would just state it formally: “the construction ensures dq / dp <= 2 at all points, thus [...]”


**Questions:**

This indistinguishability claim reminds one of the sample amplification framework proposed by https://arxiv.org/abs/1904.12053. It might be worth a remark if there’s any formal connection.

In the related work section, I would suggest citing this survey: https://arxiv.org/abs/1906.04280


**Limitations:**

It's addressed in the paper.

---

> ### Author Rebuttal · Authors · 2023-08-10
>
> Thank you for your positive assessment of our work and for your questions. We address your comments below.
>
> -----
> **High dimensional setting**: This is an interesting question, and would likely need the full extent of the notion of neighborhood optimality to address, going beyond the "$p$ versus $q$" counterexamples ("singleton neighborhoods") analyzed in this paper. We conjecture that, even though in the low dimensional case, the $\sigma \sqrt{\left.\log\frac{1}{\delta}\right/n}$ term in sub-Gaussian estimation error is replaced by our error function $\epsilon_{n,\delta}(p)$ for instance-by-instance bounds, the $\sqrt{\mathrm{Tr}(\Sigma)/n}$ term in high-dimensional sub-Gaussian estimation will probably remain the same. This is because the term is an expectation/constant probability phenomenon, and independent of the failure probability $\delta$. Also, in contrast to the low-dimensional setting where constructing the lower bound examples is challenging, in the high-dimensional setting we expect the main challenge to be finding and analyzing a neighborhood-optimal estimator.
>
> **"Worst case"**: thank you for pointing this out; we will add a clarification in the paper. We also point out that, even within the TCS community, the terms "worst case" and "instance optimality" can be used to refer to distributional instances, in the same sense our paper uses. See for example [VV16,VV17].
>
> **Sample amplification** (Axelrod et al.): that line of work is pursuing a rather different goal from ours. In both cases, we are trying to prove indistinguishability (which is a standard approach/argument), but:
>
> 1. in sample amplification, $q$ is essentially an output of the algorithm and not just a tool inside the analysis; the input $p$ is crucially *unknown* and only available via sample access.
>
> 2. in our paper, we crucially aim to construct a $q$ that has mean far from that of $p$ despite being indistinguishable, while the other paper has no particular restrictions on $q$.
>
> Thus while both works may use some techniques that are standard in the field (e.g. Hellinger distance as a proxy for indistinguishability), we think it is hard to formally connect the two frameworks.
>
> **Survey**: We agree that the Lugosi-Mendelson survey provides relevant context, and we will cite it.
>
> -----
>
> We also refer the reviewer to our general response, that, despite our result essentially settling instance-by-instance mean estimation (in 1-d) in the negative direction, we view our paper more as a "call to arms" than as a negative result. We believe that there is more work to be done in identifying reasonable distributional assumptions (in addition to symmetry), such that the sub-Gaussian error bound barrier can be overcome.

---

> > ### Comment · Reviewer_bzea · 2023-08-10
> >
> > Thanks for the response! I maintain my rating and recommend accept.
> >
> > It would be nice to add your conjecture about the high-dimensional case to the final version of the paper.

---

### Official Review · Reviewer_Z2th · 2023-07-03

**Soundness:** 4 excellent
**Presentation:** 4 excellent
**Contribution:** 3 good
**Rating:** 7
**Confidence:** 4

**Summary:**

This paper is about analyzing the optimal bounds for mean estimation, beyond the worst-case setting given in the recent breakthrough result of Lee and Valiant giving optimal subgaussian error rates for all distributions with unknown, finite variance. This prior work shows only that there exists a single "problematic" distribution class, for which the given error rates are best possible. Thus, a natural approach to make progress is to prove instance-dependent bounds for estimating the mean that are able to outperform the worst-case bounds at least for some distributions. The main result of this paper is that, assuming only that the distribution has finite mean, this natural approach cannot asymptotically improve over the worst-case optimal subgaussian error rates. In particular, given a distribution $p$, a failure probability $\delta$ and a number of samples $n$, the authors construct a distribution $q_{n,\delta}$ such that:
1. $q_{n,\delta}$ cannot be distinguished from $p$ with $n$ samples and success probability $1-\delta$.
2. The means of $q_{n,\delta}$ and $p$ differ by an amount that asymptotically approaches the optimal worst-case subgaussian bound.
Together, these two properties show that instance-dependent bounds in this setting offer no improvement. The paper also introduces a framework called "neighborhood optimality" which interpolates between the (too weak) notion of admissibility and the (too strong) instance-optimality.

**Strengths:**

The paper is well-written and gives and intuitive, straight-forward construction showing that it is not possible to improve on the worst-case optimal subgaussian bounds assuming only that the distribution has a finite mean. This is a valuable contribution by itself, and also has to deal with subtle technical difficulties arising in the case that the distribution has small mass very far from the mean, which none-the-less contributes significantly the the variance.

Further illustrating the utility of such a construction is the fact that follow-up work has already utilized the intuition behind it to prove improved bounds for symmetric distributions. In particular, the construction relies heavily on the ability to consider distributions with tails that are very skewed in one direction, and so symmetric distributions are a natural candidate for improved bounds guided by this construction.

**Weaknesses:**

The neighborhood-optimality framework seems interesting, but the results in the paper so far are not entirely convincing that this is the "right" way to go beyond the worst-case for mean estimation. However, the construction showing that subgaussian bounds are optimal when there is a finite, unknown mean is in my view the main contribution and quite significant on its own. So any potential limitations of the neighborhood-optimality framework are not much of drawback in my view.

**Questions:**

Minor comment:
Line 203 states that $q$ satisfies all the properties of definition $N_{n,\delta}(p)$, but as far as I can tell $N_{n,\delta}(p)$ hasn't been defined yet.

**Limitations:**

Yes.

---

> ### Author Rebuttal · Authors · 2023-08-10
>
> Thank you for your positive and in-depth review of our paper.
>
> Regarding your comment about the neighborhood optimality definition, we want to emphasize that, prior to this paper, the community would basically just use the local minimax definition. As we show in Appendix A, local minimax allows meaningless bounds which should be rejected as absurd, and it is thus reassuring that  our neighborhood optimality notion provides a principled framework with which to reject these bounds. Furthermore, for the common use case of indistinguishability between two distributions, neighborhood optimality is a stronger notion than local minimax. Thus, in this paper we are at least giving an improvement to a standard definition.
>
> As discussed by reviewer x13e, our paper gives a *flexible* framework, accommodating many different instantiations and interpretations, and our contribution to the mean estimation problem is philosophically different from much prior work.
> Even *if* neighborhood optimality doesn't turn out to be the "right" definition, we want to note that it is common for a community (e.g. in crypto or in various other parts of math) to "negotiate" definitions in a series of papers, until a good definition is ultimately found. We hope this paper at least initiates this conversation, by pointing out the shortcomings of local minimax and proposing an improved alternative.
> We welcome the continuing academic process of presenting a new definition which is then discussed by the community, and iterated on in future papers.

---

> > ### Comment · Reviewer_Z2th · 2023-08-11
> >
> > Your response makes sense. I didn't realize that people were regularly using the local minimax definition to prove things. In that case, I would view the contribution of this part of your paper to be two-fold: (1) Pointing out that a previously used definition doesn't really make sense (2) Proposing an alternative that does not have the same flaw. That is, a big part of the contribution of the neighborhood optimality definition is as an existence proof showing that a reasonable definition along these lines is possible.
> >
> > Thank you for your response. I will increase my score to 7.

---

### Official Review · Reviewer_x13e · 2023-07-05

**Soundness:** 4 excellent
**Presentation:** 4 excellent
**Contribution:** 4 excellent
**Rating:** 7
**Confidence:** 3

**Summary:**

This paper studies the fundamental task of 1-dimensional mean estimation. There has been recent interest in the community to understand “beyond worst-case” analyses of statistical problems, where guarantees are able to encompass properties of instances that may make them more tractable than worst-case instances. This work motivates and initiates such analysis for mean estimation. The main result aims to show negative evidence for this perspective: proving that for any distribution $p$ with finite mean there is another distribution $q$ where it is hard to distinguish $p$ vs $q$ and their means are separated by the sub-Gaussian rate. Moreover, in many senses $p$ and $q$ are “similar”, so this indicates hardness to beat the sub-Gaussian rate even when just designing an algorithm to perform on $p$ and “similar” distributions. This perspective is further developed by the formalization of “neighborhood optimality” and median-of-means is shown to be approximately neighborhood optimal.

**Strengths:**

It is an extremely fundamental pursuit to understand mean estimation and what nice properties of distributions permit better rates. The perspective introduced in this work of analyzing the optimal algorithms that perform on some distribution $p$ and “similar” distributions is both novel and insightful for progress in this pursuit.

While this paper mostly focuses on one (relatively well-motivated) notion of “similarity”, the landscape of analogous results clearly is dependent on what similarity notion is used. This is perhaps one of the most appealing aspects of the paper, as it gives a concrete framework within which to use notions of similarity as proxies for the properties that make distributions nicer in the beyond worst-case sense. As an example, the notion of similarity in this paper enforces $\frac{dq}{dp} \le 2$, indicating that the similar distributions which are still hard have similar moments and thus limiting how much moment-based guarantees can make distributions more tractable. On the other hand, the later work of [GLP23] (I believe) implies that analogous results cannot hold when similarity preserves both symmetry and Fisher information, highlighting how permissible similarity notions in this framework may help demarcate helpful properties.

**Weaknesses:**

Although likely outside the reasonable scope of this work, the existence of results such as [GLP23] indicate how this perspective would benefit from deeper investigations into various neighborhood functions.

From the exposition, it is not immediately clear how conducive the neighborhood optimality perspective is towards thinking about tightness without constant-factor lossiness (the paper seems to hint that proving a guarantee without such lossiness may be a reasonable-sounding future direction).

**Questions:**

Could you further explain your claim on line 70 that your paper shows introducing extra assumptions are necessary for the [GLP23] results? A naive reading of this claim feels strong in that worst-case lower bounds already show the need for additional assumptions, but perhaps the desired claim is that your work shows the assumptions must not be qualities that $q$ preserves in your transformation? Relatedly, does [GLP23] imply that it is open whether a result similar to your Theorem 2 is possible where $q$ additionally preserves (i) symmetry, or (ii) Fisher information, but it is impossible to (iii) simultaneously preserve symmetry and Fisher information?

Do you have more intuition to share regarding whether it is a reasonable goal to naturally hope for $(1+o(1), 1+o(1))$-neighborhood optimality?

As some of the motivation for neighborhood-optimality originates from failing the hardcoded estimator, are there known references that alternatively tackle this by restricting the estimator to have desirable properties (e.g., being translation invariant)?

Minor details:
* On line 380, is the mention of $\log(1-d^2_H(p,q)) \ge \frac{1}{2n} \log 4 \delta$ redundant from $q \in N_{n,\delta}(p)$?
* The equation above line 391 has an extra pair of parentheses

**Limitations:**

Limitations are well-discussed.

---

> ### Author Rebuttal · Authors · 2023-08-10
>
> Thank you for your insight on our work, and your questions. We particularly enjoyed reading your summary of the strengths of the paper.
>
> Here we respond to the questions raised in the review.
>
> **Q1**: What we meant is that, in order to beat the sub-Gaussian bound for *any* distribution, even asymptotically, we would need to introduce new assumptions that are not properties satisfied by the construction of $q$ in Theorem 2. And so the claim is not only about the worst case, but in fact essentially for *all* cases. We will clarify this in the paper.
>
> **Q1.5**: You are correct. If the construction of $q$ preserves both symmetry and Fisher information, then there can't be a Theorem 2 with a mean separation as large as the sub-Gaussian bound (even only asymptotically). The tight construction that yields a Fisher information rate lower bound, we conjecture, is probably $q$ being just $p$ shifted slightly, or some variant of that, without changing the shape from $p$ to $q$. The Fisher information rate bound comes from the parametric problem of location estimation. So far, there has been no proof of a finite-sample high-probability Fisher information rate lower bound for location estimation, as far as we know (from communication with the authors of [GLP23]). On the other hand, there is an essentially tight result if $p$ is sufficiently smooth (if $p$ is the convolution of some distribution with a small Gaussian), see [Gupta, Lee, Price, Valiant NeurIPS 2022].
>
> It is indeed unclear what happens if we require $\frac{\mathrm{d}q}{\mathrm{d}p} \le 2$ yet only 1 of symmetry or Fisher information not blowing up, whether we can still get a construction with asymptotically sub-Gaussian mean separation. We do want to point out, however, that a hypothetical construction preserving symmetry yet getting sub-Gaussian mean separation is almost by definition "unreasonable", in that the resulting neighborhood Pareto bound is larger than the upper bound given by [GLP23]. This suggests that the neighborhood choice is too weak, that the "$q$" construction did not preserve enough properties of $p$ (e.g. the Fisher information, in this case).
>
> **Q2**: We draw the distinction between the potential *existence* of such $(1+o(1), 1+o(1))$-optimal estimators, and their *analysis*.
>
> Existence should be doable: if we relax the neighborhood enough, then we get back to admissibility, and existence is trivial (hardcoded estimators are $(1+o(1))$-admissible, which is easy to prove). We don't see any good reason why a $(1+o(1))$-optimal estimator shouldn't exist for a more reasonable neighborhood structure.
>
> Analyzing such estimators seems much more challenging: we believe that the literature doesn't have the right tools yet.
> To get a tight analysis (through indistinguishability between two distributions), we need to develop tight tools to calculate the $n$-sample TV distance between a pair of distributions.
> However, the only generic tools we're aware of are *asymptotic* results coming from the "large deviations principle" such as Cramer's theorem, or non-asymptotic bounds that are off by constants in the exponent of the failure probability, even in the Gaussian case---based on the high-probability Pinsker inequality (based on KL divergence), or based on squared Hellinger distance. None of these approaches can give tight finite-sample bounds on the estimation error.
> Therefore, before we could refine our "$q$" construction and give tight analysis, we would first need to find better tools to bound the $n$-sample TV distance.
> This is probably a medium/long-term project, and not something accessible in the immediate future.
>
> Nonetheless, we emphasize that constant-factor results (such as those in this work) are meaningful, since variance can be much more than a constant factor different from Fisher information.
>
> **Q3**: We are unaware of additional references.
> Stein's paradox shows (particularly for the multidimensional case) that in general, we shouldn't take translation-equivariance for granted - the right estimator might not be translation-equivariant, depending on the objective/loss. We agree that it is interesting to continue to develop and refine new frameworks and perspectives on the mean estimation problem, and on beyond-worst-case analysis in general.

---

> > ### Comment · Reviewer_x13e · 2023-08-13
> >
> > Thank you for your detailed response. All of my questions have been properly answered.

---

### Official Review · Reviewer_iso5 · 2023-07-07

**Soundness:** 3 good
**Presentation:** 3 good
**Contribution:** 3 good
**Rating:** 6
**Confidence:** 3

**Summary:**

This paper studies the 1-dimensional robust mean estimation from the perspective of "beyond the worst-case analysis". They show somehow a negative result, that is, for any distribution, we can always construct another distribution that is close in probability distance, but their means are well-separated. Furthermore, to analyze the fine-grained optimality of algorithms, they propose a new definitional framework called “neighborhood optimality”. They show that the classic MoM estimator is neighborhood optimal up to some constant.

**Strengths:**

- This paper studies the 1-dimensional robust mean estimation from the perspective of "beyond the worst-case analysis", which is a very important and interesting view for this problem.

- Their construction might be adapted in other settings.

**Weaknesses:**

- As discussed in the paragraph (lines 69-76), there exists a class of distributions (symmetric distributions) whose error rate can be better than the sub-Gaussian rate. My question is can we modify the current construction to reflect this result by restricting the feasible domain of the perturbed distributions?

**Questions:**

N/A

---

> ### Author Rebuttal · Authors · 2023-08-10
>
> Yes, one could use the framework of this paper to attempt to design a construction for symmetric distributions $p$ (constructing a symmetric alternative, $q$), and thereby obtain a Fisher information rate estimation lower bound. In fact, (as discussed in [GLP23]) the Fisher information rate bound of symmetric mean estimation really comes from the related problem of location estimation, which is the "parametric" version of mean estimation: suppose we know the density of the data distribution, but not its shift, the goal is to estimate the shift from i.i.d. data. As such, the correct lower bound construction is likely just shifting $p$ slightly to form $q$, or slight variants of this.
>
> The algorithmic techniques of [GLP23] for symmetric mean estimation come from their study of the location estimation problem in previous papers. From our communication with the authors, they have also been trying to prove lower bounds for location estimation, but so far they have not been successful in getting a finite-sample and high-probability Fisher information rate bound. Thus, it remains an open question to show such a bound. On the other hand, if we only care about the error variance, then the well-known Cramer-Rao bound states that every shift-invariant location estimator must have variance at least $I/n$, where $I$ is the Fisher information of the distribution. So there's already good (but not definitive) evidence that the Fisher information rate is optimal.
>
> We also refer the reviewer to our general response discussing how we view our paper more as a "call to arms" than as a negative result. We believe that there is more work to be done in identifying reasonable distributional assumptions (besides symmetry), such that the sub-Gaussian error bound barrier can be overcome.

---

> > ### Comment · Reviewer_iso5 · 2023-08-19
> > **Thanks for your reply**
> >
> > Thanks for your detailed response, which I have learned a lot. I also agree that such an attempt o beyond-worst-case analysis is very meaningful. Hence, I decide to increase my score from 5 to 6.

---

### Author Rebuttal · Authors · 2023-08-10

We thank the reviewers for the positive assessment of our work, and of course also for the constructive reviews and many interesting questions raised.
We are particularly encouraged that the reviewers find our paper 1) well-motivated/important (Reviewers iso5, bzea), 2) an extremely fundamental pursuit (Reviewer x13e), 3) giving a surprising result (Reviewer bzea), 4) significant (Reviewer Z2th) and 5) ultimately, opening up a new research direction (paraphrasing Reviewer x13e).
Here, in this overall response, we would like to re-emphasize the main conceptual message and contribution of the paper in light of the reviews.
We address the research questions raised by the reviewers as responses to each review.

This paper studies the problem of optimal mean estimation in the most basic and widely used setting of 1 dimension. We go beyond worst case analysis, by considering optimality on an instance-by-instance basis. Motivating this from a different perspective:  Gaussians are known to be hardest instances for mean estimation under the finite but unknown variance assumption. We thus ask, are there "easier" distributions for which algorithms can beat the sub-Gaussian error bound? Can an algorithm leverage the beneficial structure of a distribution, without explicit knowledge of this structure?

Our paper provides an unexpected and subtle answer: "yes in limited cases/regimes, but in general no". For some distributions, even standard algorithms can beat the sub-Gaussian bound, but only for a limited parameter regime *per distribution*---namely, if the number of samples is not too large (Proposition 14). In general however, we show a strong and comprehensive negative result. Fixing any distribution $p$, then for large enough number of samples $n$: no estimator can outperform the sub-Gaussian bound by more than a constant factor, unless it has essentially "hardcoded" knowledge of $p$, and thereby misestimates another "similar" distribution $q$ (see Theorem 2/Corollary 3, and the construction of $q$ in Definition 4).

As reviewer x13e puts it, our paper provides a correspondence, mapping a neighborhood structure---"similarity" notion---to a beyond-worst-case analysis. For the natural and minimalist neighborhood notion that "$q$ has tails not much larger than those of $p$", we show a strong negative result, essentially ruling out better-than-Gaussian performance. The key point---a **call to arms** in a sense---is that such negative results are to be circumvented, through identifying additional favorable distribution structure for the mean estimation problem.
Inspired by our results and construction, [GLP23] already showed that if we additionally assume the data distribution to be *symmetric* about its mean, then it is possible to get substantially (and in fact arbitrarily) better-than-Gaussian performance, in fact achieving the Fisher information rate.

We hope the perspective and framework introduced in this paper will inspire further work, investigating structures other than symmetry, towards understanding how to break worst-case performance barriers and overcome them algorithmically.

---

### Decision · Program_Chairs · 2023-09-21

**Decision:**

Accept (poster)

**Comment:**

This paper studies optimal bounds for mean estimation in one-dimensional distributions, beyond their worst-case settings. The existing work has already characterized the optimal Subgaussian error rates for all one-dimensional distributions with unknown, but finite variance. This paper strives to answer the following question: Are there instance-dependent bounds for estimating the mean that are able to outperform the worst-case Subgaussian error rates? This paper provides a strong negative answer to this question. In particular, given a probability distribution with an unknown but finite mean, this paper designs a surrogate distribution that satisfies two key properties: (1) the two distributions cannot be distinguished with a fixed number of samples and a fixed probability, (2) The means of these distributions differ by at most the aforementioned Subgaussian error rates.  Further, the paper introduces a new framework to analyze the fine-grained optimality of algorithms, which the authors call “neighborhood optimality”. As an application of this new framework, the authors show that the standard median-of-means algorithm is neighborhood optimal, up to constant factors.

Four reviewers have reviewed the paper, and their overall assessment of the paper was very positive. I agree with this assessment and believe that the paper has important and interesting implications within the field of robust mean estimation.

The authors are highly encouraged to take into consideration the comments made by the reviewers. The AC also suggests that the authors include a brief discussion on the possible extension of their approach to the robust mean estimation in higher dimensions (especially the interplay between the dimension and the error rate).